# FLASHIda enables intelligent data acquisition for top-down proteomics to boost proteoform identification counts

Kyowon Jeong [1,2 ✉], Maša Babović [3], Vladimir Gorshkov [3], Jihyung Kim[1,2], Ole N. Jensen [3] & Oliver Kohlbacher [1,2,4 ✉]

The detailed analysis and structural characterization of proteoforms by top-down proteomics (TDP) has gained a lot of interest in biomedical research. Data-dependent acquisition (DDA) of intact proteins is non-trivial due to the diversity and complexity of proteoforms. Dedicated acquisition methods thus have the potential to greatly improve TDP. Here, we present FLASHIda, an intelligent online data acquisition algorithm for TDP that ensures the real-time selection of high-quality precursors of diverse proteoforms. FLASHIda combines fast charge deconvolution algorithms and machine learning-based quality assessment for optimal precursor selection. In an analysis of *E. coli* lysate, FLASHIda increases the number of unique proteoform level identifications from 800 to 1500 or generates a near-identical number of identifications in one third of the instrument time when compared to standard DDA mode. Furthermore, FLASHIda enables sensitive mapping of post-translational modifications and detection of chemical adducts. As a software extension module to the instrument, FLASHIda can be readily adopted for TDP studies of complex samples to enhance proteoform identification rates.

[1] Applied Bioinformatics, Computer Science Department, University of Tübingen, Sand 14, 72076 Tübingen, Germany. [2] Institute for Bioinformatics and Medical Informatics, University of Tübingen, Sand 14, 72076 Tübingen, Germany. [3] Department of Biochemistry & Molecular Biology and VILLUM Center for Bioanalytical Sciences, University of Southern Denmark, Campusvej 55, DK-5230 Odense M, Odense, Denmark. [4] Translational Bioinformatics, University Hospital Tübingen, Hoppe-Seyler-Str. 9, 72076 Tübingen, Germany. ✉email: kyowon.jeong@uni-tuebingen.de; oliver.kohlbacher@uni-tuebingen.de

Top–down proteomics (TDP) enables comprehensive and in-depth analysis of intact proteoforms (i.e., protein species arising from the same gene product via splice variants, genomic variation, post-translational modifications, degradation, etc.[1–3]). Proteoforms have high phenotypic heterogeneity in different biological systems, and proteoform level information can provide important insights into biochemical function or disease phenotypes[4–8]. TDP as the method of the choice to study proteoforms has thus become important for many biomedical applications[9,10].

In recent years, significant improvements have been made to sample processing, separation, fragmentation, and bioinformatics methods for TDP[11–15]. As a result, proteoforms that have been difficult to analyze by TDP (e.g., proteoforms of large masses and of membrane proteins) have become easier to detect and characterize[8,9]. In large-scale studies of complex samples such as microbial or human cell lysates, the number of proteoform identification has been increased up to 4000–6000 (corresponding to 500–1000 proteins)[16,17]. In single-shot TDP experiments, about 800 proteoforms could be identified in *E. coli* lysate[18] and about 1800 proteoforms in a human brain sample[19]. The broad coverage in these experiments can be attributed to technical improvements like capillary zone electrophoresis (CZE) and high-field asymmetric waveform ion mobility spectrometry (FAIMS)[19–23] as well as the state-of-the-art bioinformatics analysis tools like TopPIC[14] and ProSight PC (Thermo Fisher).

At the same time, fragmentation strategies implemented in data-dependent acquisition (DDA) by current instrument software have been optimized to support bottom-up proteomics (BUP) rather than for TDP. These strategies usually select $N$ most intense peaks (typically, $N$ is in a range of 10–20) for fragmentation acquisition, often excluding previously selected ones over a short retention time (RT) period (Top-$N$ acquisition with a dynamic exclusion list)[24]. While this scheme effectively captures diverse peptide ions of high quality in BUP studies[25], these selection criteria are suboptimal for high-quality proteoform ion selection in TDP. In contrast to peptide ions in BUP, a single proteoform generates many peaks due to its large mass and high charges. Top-$N$ acquisition thus often results in the selection of multiple peaks from a single abundant proteoform rather than from multiple distinct proteoforms, which in turn results in poor proteoform coverage. In addition, the exclusion list preventing the selection of the same m/z (mass-to-charge ratio) region does not guarantee the exclusion of the same proteoform (mass). Also, the intensity based exclusion list may not lead to the selection of high quality precursors that generate many unique fragments.

Most large-scale TDP studies, however, use DDA acquisition with specifically tuned parameters, for instance, relatively low $N$ values (ranging from 3 to 5) for Top-$N$ acquisition and relatively high isolation window size ranging from 1.2 to even 15 Th (ultra-wide isolation)[6,18]. Post hoc analysis of the selected precursor ions shows that the selection of the proteoforms is far from ideal. Smarter acquisitions, sometimes termed 'intelligent data acquisition' (IDAs), have thus been discussed in the literature (e.g., Durbin et al.[26] and Lu et al.[12]). Both methods used real-time mass deconvolution (i.e., real-time determination of intact proteoform masses) to disentangle complex signal structure of proteoforms. Autopilot[26–28] also employed real-time database search for better characterization of proteoforms as well as for less redundant precursor mass selection, while MetaDrive[12] focused on improving the quality of the fragment spectrum by dissociating multiple precursor ions of the same proteoform. Both approaches clearly demonstrated the potential of IDA for TDP studies and, at the same time, were limited because of the need for faster processing times that could fit within the cycle time of the MS instrument.

Here, we present FLASHIda, a machine learning-based IDA algorithm designed to maximize proteoform coverage in TDP. FLASHIda interfaces with tribrid Thermo Scientific mass spectrometers through the instrument API (iAPI) allowing for real-time access to MS data. By combining our recently developed real-time spectrum deconvolution algorithms[29] and a machine learning technique assessing the quality of the precursor isotopomer envelopes, FLASHIda enables non-redundant selection of precursor ions of very high quality during the LC-MS run and thus boosts the proteoform coverage.

## Results

**FLASHIda overview**. Figure 1a illustrates the MS duty cycle control employed by FLASHIda. FLASHIda processes each MS full scan within a few milliseconds (about 20 ms on average) and optimizes the acquisition of the next cycle to maximize isoform diversity in acquisition (see Methods for the algorithm). Figure 1b illustrates the key steps of FLASHIda. FLASHIda uses Thermo iAPI to access MS full scan in real-time. FLASHIda takes three steps to select high quality precursor isotopomer envelopes (simply precursors from here on) of diverse proteoforms. The first step is to transform the input m/z-intensity spectrum into a mass-quality spectrum, and the second is to select precursors in the transformed spectrum so that the number of uniquely identified proteoform masses is maximized. Lastly, the charge state and isolation window size for each selected mass are dynamically determined to minimize interference from noise or coelution. The determined isolation window m/z ranges are provided to the instrument through the Thermo iAPI interface.

**Spectrum deconvolution and quality prediction**. On receiving MS1 full scan, FLASHIda performs mass deconvolution within 10–50 ms based on the fast decharging algorithm developed for FLASHDeconv[29]. Since a single (monoisotopic) mass is represented by multiple isotopomer envelopes of distinct charges, this transformation drastically reduces the signal complexity, already making the selection of distinct proteoform masses far simpler than in the original spectrum.

We then employ a machine learning model to predict the quality of the resulting fragmentation (Fig. 1c). Based on six relevant features (see below and Methods) extracted from the original mass spectrum, we have trained a logistic regression model to compute a quality metric (termed QScore), which predicts a probability for the resulting fragment spectrum to be successfully identified. Note that this does not require an actual identification (as in AutoPilot), but merely estimates a likelihood for success based on the observation that higher 'quality' of the precursors affects the quality of the fragment spectrum and thus the identification likelihood. In preliminary studies, we found that the following features are easy to extract (speed is important for the online processing) and still contain sufficient information to assess the precursor quality: the shape of isotopomer envelopes (as compared to theoretical ones), signal-to-noise ratio (SNR) within the precursor envelope range, intensity distribution over different charges, and average mass errors of all peaks. The first two features (the isotopomer envelope shape and SNR) are defined both for peaks within precursor ion m/z range and for peaks within precursor ion mass range (of all possible charges). Similar features were used to measure deconvolution quality in Marty et al.[30]. Detailed regression procedure and definition of the features are given in Methods. The calculated QScores of precursor masses can then be used to prioritize precursor selection in a manner that favors precursors with better odds of identification.

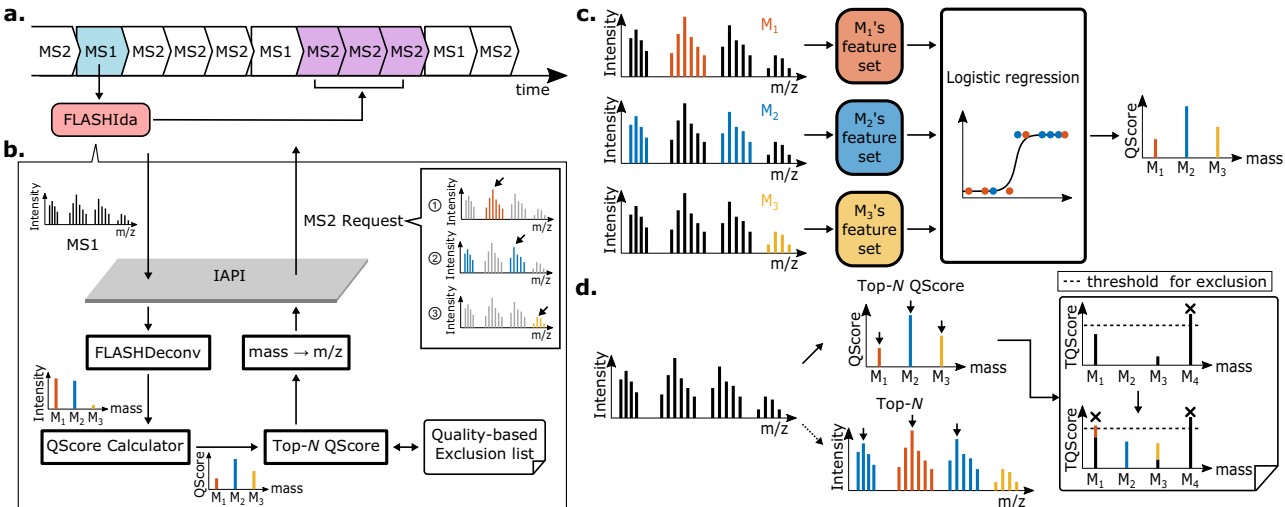

**Fig. 1 Overview of FLASHIda. a** MS duty cycle control employed by FLASHIda. FLASHIda processes each MS full scan within a few milliseconds (about 20 ms on average) and optimizes the acquisition of the next cycle. **b** Key steps of FLASHIda. FLASHIda uses Thermo iAPI to access MS full scan in real-time. FLASHIda takes three steps to select high quality precursor isotopomer envelopes of diverse proteoforms. The first step is to transform the input m/z-intensity spectrum into a mass-quality (QScore) spectrum, and the second is to select precursors in the transformed spectrum so that the number of unique proteoform level identifications (simply proteoform ID) is maximized (Top-N QScore precursor acquisition with a quality-based exclusion list). Lastly, the charge state and isolation window size for each selected mass are dynamically determined to minimize interference from noise or coelution. The determined isolation window m/z ranges are provided to the instrument through the Thermo iAPI interface. **c** QScore calculation via logistic regression. After the deconvolution, the quality of the resulting fragmentation is measured for each precursor using a logistic regression with six features (see Methods). The features are extracted from the peaks in the original spectrum corresponding to the precursor. The quality metric, or QScore, is the probability estimated by the regression that the resulting fragment spectrum to be successfully identified. **d** Top-N QScore precursor selection with a quality-based exclusion list. With the transformed mass-QScore spectrum, FLASHIda attempts to select the high quality precursors while maximizing the identified proteoform diversity, simply by taking the top N masses with the highest QScore (Top-N QScore acquisition). To reduce the redundancy in precursor ion selection, FLASHIda conservatively excludes masses that are highly likely to be already identified; if a mass has been acquired multiple times within a short RT duration, the probability that at least one of the acquired MS2 spectra from the mass is identified is calculated with its QScores (see Methods for detail). This probability, called TQScore, is immediately updated per mass upon each acquisition, and the masses of high TQScores (>0.9) are excluded from the acquisition for a short RT duration.

**Precursor prioritization.** Once the quality scores of precursors have been computed, FLASHIda selects the best precursors from all available masses. To maximize the proteoform diversity, high-quality precursors from yet unfragmented proteoforms should be selected. The selection of high-quality precursors can be readily done based on the QScore of the precursors, simply by taking the top N masses with the highest QScore that have not yet been selected (Fig. 1d). In order to avoid re-fragmentation of the same proteoforms, we apply dynamic exclusion (inspired by the dynamic exclusion used in BUP) to maximize the diversity of the selected proteoforms.

We found that the number of high quality proteoform precursors at a given RT is often very low as compared with that of peptide precursors in BUP. Therefore, the use of the naive mass exclusion alone resulted in the selection of low quality masses, and in turn an overall drop of identification rate (see Supplementary Fig. 1). Similar to the exclusion by the confidence of the identification used in Autopilot, FLASHIda only excludes masses that are highly likely to be already identified. To this end it keeps a list of the triggered masses over a short RT duration along with their QScores. If a mass has been acquired multiple times, the probability that at least one of the acquired MS2 spectra from the mass is identified is calculated with its QScores (see Methods for details). Note that this probability, called TQScore (Total QScore) of the mass, is the probability that the proteoform of the mass is identified. After each acquisition, TQScores are immediately updated. If the TQScore of a mass exceeds 0.9, the mass is registered in the exclusion list for a short RT duration. If all deconvolved precursor masses are already registered in the

exclusion list, FLASHIda selects the one with the lowest TQScore. Using this quality-based mass exclusion list, FLASHIda assures each triggered mass (of proteoform ion) is identified while still maximizing the diversity of proteoforms.

**IDA doubles the number of proteoform IDs compared to DDA.** To benchmark FLASHIda, we generated four sets of nano-RPLC MS/MS single-runs from *E. coli* lysate: two sets using FLASHIda acquisition, and two using standard acquisition for intact proteins (Intact Protein Mode) on a Thermo Scientific Orbitrap Eclipse (see Methods for details). One set from each acquisition has a gradient time of 30 min and another 90 min. Each set was measured in technical triplicate, thus a total twelve measurements were obtained. The triplicates of 90 and 30 min FLASHIda runs are collectively denoted as FI90s and FI30s, respectively. Likewise, the standard runs are ST90s and ST30s. FI90s and FI30s are collectively called FI, and likewise ST. We note that the datasets used to train QScore are not part of this benchmark data.

In the benchmark tests, we distinguish proteoform level identifications, or simply proteoform IDs, and biological proteoforms since multiple proteoform IDs in the search results may represent simple chemical adducts of the same biological proteoform.

For acquisition (both FLASHIda and standard), we used Top-4 acquisition and set precursor charge range between 4 and 50. For standard, dynamic exclusion with 30 s exclusion duration was used. For standard, a fixed isolation window size of 3.0 Th was

used for all runs. For FLASHIda runs, the isolation window size is dynamically controlled so the isolation window covers the complete precursor envelope range with 0.6 Th margin from both sides. On average, the isolation window size from FI datasets was 3.1 Th.

For FLASHIda, mass range was set to from 400 to 50,000 Da, and the TQScore threshold was 0.9. In addition, FLASHIda discarded any precursors with QScore lower than 0.25 so only high quality precursors are triggered. Lastly, FLASHIda also discarded precursors of SNR within the precursor envelope m/z range (called precursor SNR; see Methods) lower than 1.0. Briefly, precursor SNR is defined by the power of targeted peaks divided by that of non-targeted peaks within the precursor envelope m/z range. When coeluted ions are present within the m/z range (resulting in so-called chimeric spectra) or the precursor mass represents mass artifacts (see examples in Supplementary Fig. 2–6), the power of the non-targeted peaks increases and in turn the precursor SNR decreases. Since the chimeric spectra and the spectra of incorrect precursor masses often cause target-decoy based false discovery rate (FDR) control inaccuracies and even erroneously inflate the number of proteoform IDs (see Methods, Supplementary Figs. 7, 8 and Supplementary Data 1 for detail), the precursors of low precursor SNR were avoided in data acquisition.

For data analysis, the twelve LC-MS/MS datasets were analyzed in the same way: raw files were converted into mzML files[31] using ProteoWizard[32] msconvert (version 3.0.20186-dd907d75; see Supplementary Fig. 9 for GUI screenshot). Both MS1 and MS2 spectra were deconvolved by FLASHDeconv (version 2.0 beta), and identified by TopPIC (version 1.4.4). For FLASHDeconv, charge range was set to a range from 4 to 50 and mass range from 400 and 50,000 Da (identical with FLASHIda acquisition parameters). Then, as for the acquisition, the precursor SNR filtration was applied for all deconvolved precursor masses with the threshold of 1.0; this threshold was already applied for FI datasets by FLASHIda but not for ST datasets. For TopPIC, one unknown modification was allowed for search, and the FDR threshold was set to 1% (both in spectrum and proteoform levels). We used the E. coli (strain: K12 MG1655 i) database in fasta format downloaded from SwissProt (downloaded 28.05.2020; Supplementary Data 2). The FLASHDeconv and TopPIC commands are shown in Methods, and the identification results from FI and ST datasets are provided in Supplementary Data 3, 4.

Figure 2a shows the number of unique proteoform IDs, proteins, and spectra. FI90s resulted in nearly 1500 proteoform IDs on average, almost doubling the number of proteoform IDs found in SI90s (792 on average). From FI30s, on average 764 proteoform IDs have been found, which is comparable to SI90s although FI30s used only a third of the machine time compared to SI90s. SI30s yielded the minimum proteoform ID count of 422 on average. In terms of protein count, FI90s reported 338 average protein count while SI90s reported 259. FI30s also reported 247 proteins on average, again demonstrating the high performance of FLASHIda in short gradient runs. ST30s reported about 181 proteins. The right panel of Fig. 2a compares the number of acquired, deconvolved, precursor SNR threshold applied, and identified precursors (or MS2 spectra). The numbers of identified MS2 spectra in FI datasets were almost twice as many as in ST, resulting in a rate of ~75% of fragment spectra being identifiable in FLASHIda runs, in contrast to ~35% using DDA. While the ST datasets have almost 20% more precursors acquired than in the FI dataset (40,643 in ST vs. 32,997 in FI), the number of deconvolved precursors in ST datasets are almost 30% fewer than in FI (25,579 in ST vs. 32,702 in FI). The precursors of high precursor SNRs were more than twice as large in the FI than in the ST dataset (14,255 in ST vs. 32,208 in FI). Out of these

precursors, more than twice as many distinct nominal masses were acquired for FI than for ST (1924 in ST vs. 4592 in FI), demonstrating that FLASHIda results in precursors of far diverse masses and of higher quality than standard DDA.

When we compared the proteoform IDs from each acquisition, 1774 nominal proteoform ID masses were exclusively identified in FI datasets, and only 105 (5.9%) masses out of these exclusive masses were acquired and passed precursor SNR filtration (but not identified) in the ST datasets. On the other hand, out of 217 exclusive nominal proteoform ID masses in the ST datasets, 100 (46.1%) were acquired but not identified in the FI datasets. Per each proteoform ID, the average numbers of acquired precursors were 4.8 for FI and 7.0 for ST, demonstrating a less redundant precursor selection of FLASHIda. In contrast, the average numbers of identified MS2 spectra per proteoform ID were 3.5 for FI and 3.2 for ST; thus, FLASHIda acquired fewer spectra, but identified more MS2 spectra per proteoform ID than standard DDA.

To see if the increased unique precursor mass count is due to the frequent one Da mass error arising from inaccurate deisotoping in precursor mass deconvolution[29], we examined the mass difference between deconvolved precursor mass and proteoform mass determined by identification (with Precursor mass and Adjusted precursor mass columns in TopPIC output tsv files). Standard DDA and FLASHIda generated comparable ratios of one Da mass error (7.2% and 9.9% in ST90s and FI90s datasets, respectively). Moreover, for heavy proteoform IDs (>25 kDa), FLASHIda showed less error ratio than standard DDA (see Supplementary Figs. 10, 11).

We then investigated the quality of MS2 spectra that are commonly identified by FLASHIda and standard DDA. To this end, from FI and ST datasets, we collected 1171 jointly identified proteoform IDs. Then for each dataset, we took 1171 MS2 spectra representing the collected proteoform IDs (found in *proteoform.tsv files reported by TopPIC). For these MS2 spectra, the average value of $-\log_{10}$(evalue) was higher for those from FI than for ST (11.34 vs. 11.05; the higher the better), suggesting the MS2 spectra by FLASHIda have better quality in terms of the database match score. Moreover, when we compared the ambiguity in the position of modification on proteoform ID reported by TopPIC, it was lower for FI spectra than ST (17.3 vs. 18.5 on average for proteoform IDs with modifications). These results show that FLASHIda successfully selects high quality precursors from diverse proteoform IDs, significantly increasing the number of proteoform IDs (see also Supplementary Figs. 12–14 for more spectrum level analyses).

Next, we assessed the prediction accuracy of the precursor quality score. Figure 2b shows the histograms of identified and unidentified spectra with respect to QScore (left panel) and precursor intensity (right) from the ST (top) and FI (bottom) datasets. The QScore distributions from the ST datasets demonstrate that QScore separates the identified and unidentified ones far better than intensity. A large portion of high intensity precursors in the ST datasets were not identified while most high QScore precursors were identified. In addition, it is shown that QScore predicts the identification rate accurately. In FI datasets, the histogram is truncated and does not show low QScores, as precursors with values below 0.25 are discarded in IDA. Also the identification rate is underestimated, which should be because QScore was used for the acquisition rendering the prediction biased (QScore was trained on standard DDA acquired data). Comparing precursor intensity distributions (Fig. 2b, right panel) reveals that FI precursors have lower intensities than ST precursors on average. This is because precursor prioritization in the standard DDA acquisition is mostly intensity-based while that in FLASHIda is quality (QScore)-based. However, FLASHIda

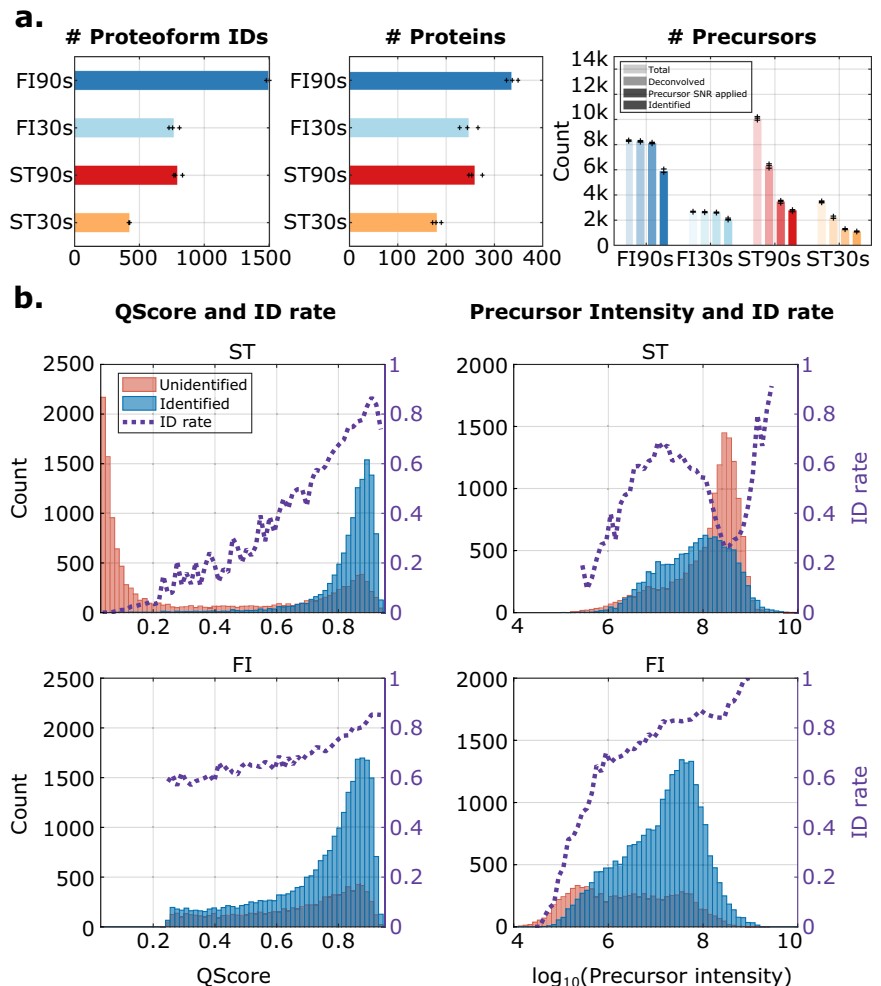

**Fig. 2 FLASHIda doubles the number of proteoform IDs compared to standard DDA. a** Identification results from FLASHIda as compared with the standard acquisition. $n = 3$ technical replicates were examined over four independent single LC-MS/MS runs from *E. coli* lysate sample. The four sets of *E. coli* lysate single runs were 90 min gradient technical triplicates with FLASHIda acquisition (FI90s), 30 min FLASHIda (FI30s), 90 min standard acquisition (ST90s), and 30 min standard (ST30s). Top-4 acquisitions were used for both methods. Each dataset was deconvolved by FLASHDeconv and identified by TopPIC at 1% spectrum and proteoform level FDR (see Methods). The left panel shows the unique proteoform ID count from each set, and the middle panel the protein count. The right panel shows the numbers of total triggered, deconvolved, precursor SNR applied, and identified (from lightest to darkest in each grouped bars) precursors from each set. The (+) markers show the numbers from the replicates, and the bars show their average values. It is clearly shown that FLASHIda reports almost twice more unique proteoform IDs than the standard acquisition, or alternatively similar numbers as with standard acquisition on drastically shorter gradient runs. In terms of protein count, almost 30% increase was observed. The spectrum level analysis in the right panel shows that FLASHIda runs result in far better identification rates (~75%) than standard (~35%). Source data are provided as a Source Data file. **b** The histograms for identified (blue) and unidentified (red) spectra over QScores (left panel) and precursor intensities (right) from the standard acquisition (ST, top panel) and FLASHIda (FI, bottom). For each plot, the identification rate (ID rate) in each bin is also shown as a purple dotted line. For FLASHIda runs, the distribution over QScore is truncated as precursors of QScore less than 0.25 were discarded. The top panel (ST) shows that QScore not only far better separates between identified and unidentified spectra than precursor intensity but also estimates the identification rate well in standard runs. In FLASHIda runs, QScore is underestimating the identification rate, which should be because QScore was used for the acquisition making the prediction biased. Comparing precursor intensity distributions also reveals that FLASHIda achieves almost an order of magnitude wider dynamic range than the standard acquisition. Source data are provided as a Source Data file.

achieves almost one order of magnitude higher dynamic range than standard DDA, suggesting that FLASHIda enables the selection of low-intensity, but high-quality precursors.

The advantage of FLASHIda acquisition is maximized when coupled with FLASHDeconv in the analysis because the precursor masses from both tools are consistent with each other. When the FI and ST datasets were deconvolved by TopFD[14] (for both MS1 and MS2) instead of FLASHDeconv, only 5–10% more proteoform IDs were found in FI datasets than in ST datasets (Supplementary Fig. 15a and Supplementary Data 5–9). We also found that TopFD yields more proteoform IDs from ST datasets

than FLASHDeconv. When the precursor SNRs of TopFD proteoform IDs in ST datasets were measured, about 30% of them had the SNRs lower than 1.0, which contributed to the increased proteoform IDs (as described above, FLASHDeconv only retains proteoforms of precursor SNRs higher than 1.0). Thus, we attributed the low ID boost in FI datasets to these low precursor SNR proteoforms and applied the precursor SNR threshold of 1.0 to all TopFD IDs (see Supplementary Figs. 16–20 for examples of low precursor SNR TopFD IDs). This filtration, as expected, removed 30% of TopFD IDs from ST datasets. But we also observed that 30% of the proteoform IDs from FI datasets

were removed by this filtration, again yielding less than 10% improvements on proteoform ID on FI datasets. (Supplementary Fig. 15b). These removed spectra in FI datasets obviously have high precursor SNR ( > 1.0) when FLASHDeconv (in FLASHIda) is used for deconvolution, since FLASHIda acquired only such precursors. But they have low SNR ( < 1.0) when TopFD is used instead. This discrepancy arises because precursor SNR is determined by not only the precursor but also its deconvolved mass; when different deconvolution methods report different masses, a single precursor may have different precursor SNRs. In the opposite case, there may be precursors of high SNR with TopFD deconvolution but of low SNR with FLASHDeconv. These are typically not acquired by FLASHIda. Thus, using FLASHdeconv's mass deconvolution for FLASHIda acquisition ensures that the acquired precursors have high SNR values leading to high identification rate.

Indeed, when the precursor masses from FLASHIda were used instead of TopFD precursor masses (with MS2 deconvolution still done by TopFD), the proteoform ID counts for ST datasets almost did not change, but those for the FI datasets increased by 50–65%, recovering the boost by FLASHIda acquisition (see Supplementary Fig. 15c and Supplementary Data 10; note that in Figs. 2 and 3, FLASHDeconv was used for both MS1 and MS2 deconvolution). Since MS2 deconvolution is usually dependent on precursor mass and charge values, the boost could be even higher if an interface between FLASHIda and TopFD were implemented.

**IDA increases the number of overlapping proteoform level identifications**. Then we measured the proteoform and protein level reproducibility across all FI and ST datasets. Two proteoform IDs (from different datasets) are defined to match each other if they have the same amino acid sequence and masses within 1.2 Da tolerance. The overlap coefficients (the size of the intersection divided by the smaller of the size of two compared sets) between the datasets are given in Fig. 3a. Even with significantly larger numbers of proteoform IDs and proteins than in the ST datasets, the FI datasets showed comparable overlap coefficients with ST datasets, almost doubling the overlapping proteoform IDs between replicates. For both datasets, proteoform level reproducibility (0.5–0.8) was rather lower than protein level (~0.8). As seen in Supplementary Fig. 21, proteoform reproducibility is more strongly correlated with precursor intensity than with quality (QScore). We thus presume that we are hitting the limits of stochasticity of detectability, which renders the low proteoform level reproducibility. Indeed, in many cases more than tens of proteoform IDs from a single gene (or protein) were identified with a high dynamic range. For example, 40 proteoform IDs were reported from DNA-binding protein H-NS (UniProtKB: P0ACF8) in a single FI90 dataset with a dynamic range of four orders of magnitude. Thus, for an overlapping protein, often some of its low abundant proteoform IDs were not detected across all datasets.

**Analysis of proteoform and protein coverage**. We then analyzed the proteoform IDs reported in the FI90s datasets. All 4,483 proteoform IDs are shown on the scatter plot on the RT-mass plane in Fig. 3b. The deconvolved mass signal is shown to be rather sparse in particular in the later part of the gradient (60–80 min) where larger proteoforms elute. This would be because of low ionization efficiency and/or low signal (mainly consisting of isotopically unresolved peaks) quality of heavy proteoforms (>30 kDa). However, as compared to other *E. coli* TDP studies, far more heavy proteoform IDs were identified. For example, in a single FI90 dataset, 105 proteoform IDs with mass

larger than 30 kDa have been identified (7% out of all proteoform IDs) while only 52 were identified (0.1% out of all) in a large-scale TDP analysis done by McCool et al.[17] (2D LC separation, 43 CZE-MS/MS runs). In the ST90 dataset, only 35 proteoform IDs had masses larger than 30 kDa. The dynamic range of the proteoform intensities (measured by the feature area of each proteoform ID; different from precursor intensity dynamic range in Fig. 2b) was about five to six orders of magnitude for both FI90s and ST90s datasets, but FI90s reported far more proteoform IDs with lower proteoform intensities, showing the proteoform ID count boost by FLASHIda is mostly achieved for low abundance proteoform species (see Supplementary Fig. 22).

Then, we compared the identified proteins in the FI90s and ST90s datasets with published quantitative BUP data for *E. coli* MG1655 cells grown in LB medium from the study by Schmidt et al.[33]. From FI90s, 245 proteins jointly detected in triplicates were collected. Likewise, 182 proteins were collected from ST90s triplicates. Out of the 50 proteins reported to have the highest copy number per cell, 42 and 40 were identified in FI90s and in ST90s datasets, respectively (Fig. 3c). Six of the eight undetected proteins were heavy proteins (>35 kDa; see Supplementary Data 11). Figure 3c also shows that FI90s datasets had 63 exclusive jointly detected proteins while ST90s had only two.

We performed a gene ontology (GO) term analysis for the identified proteins, using the Gene Ontology Resource[34,35] for both FI90s and ST90s datasets. The annotations were done with "PANTHER GO-Slim" datasets, and binomial tests at 5% false positive rates were used. For FI90s datasets, the cellular component analysis (in Fig. 3d; also see Supplementary Data 12) shows that most abundant proteins are ribosomal proteins, as reported in Ishihama et al.[36], another *E. coli* study done by BUP. On the proteoform level as well, almost 40% of identifications were from ribosomal proteins: 28% from 50 S and 12% from 30 S subunits. However, the number of proteoform IDs per ribosomal protein varied widely; for example, while only seven unmodified proteoform IDs of 50 S ribosomal protein L15 (UniProtKB: P02413) were reported, 34 proteoform IDs of 50 S ribosomal protein L31 type B (UniProtKB: P0A7N1) were identified in a single FI90 dataset. The relative intensities of the ribosomal proteoform IDs from the same protein were also often two orders of magnitude apart (e.g., UniProtKB: P0A7N1, P0A7K2, and P0A7N9).

In addition to ribosomal proteins, various proteins from distinct components including intracellular organelles, cytosol, and cytoplasm were identified in FI90s datasets. Not surprisingly, membrane proteins were rarely reported, indicating that analyzing intact membrane proteins via TDP is still difficult when using standard conditions. In ST90s dataset, almost identical GO terms were found as in FI90s, in particular for ribosomal proteins. However, no membrane protein related GO terms were reported from ST90s dataset.

In the biological process analyses (in Supplementary Data 12) for the FI90s datasets, the most predominant GO term was *cytoplasmic translation*. Proteins involved in ribosomal subunit assembly were reported as abundant, and proteins associated with energy-related GO terms were also found to be abundant (*ATP generation*, *ADP metabolic process*). In terms of molecular function (in Supplementary Data 12), many proteins with binding functions including mRNA, rRNA, and ribosome binding were found. These results are highly consistent with the results from quantitative BUP analysis in Ishihama et al.[36]. In ST90s datasets, however, far fewer biological process GO terms were reported (96 in FI90s vs. 59 in ST90s), and the energy-related terms were not reported. The molecular function GO analysis results from FI90s and ST90s were almost identical. We can thus confidently state that FLASHIda is able to provide rich

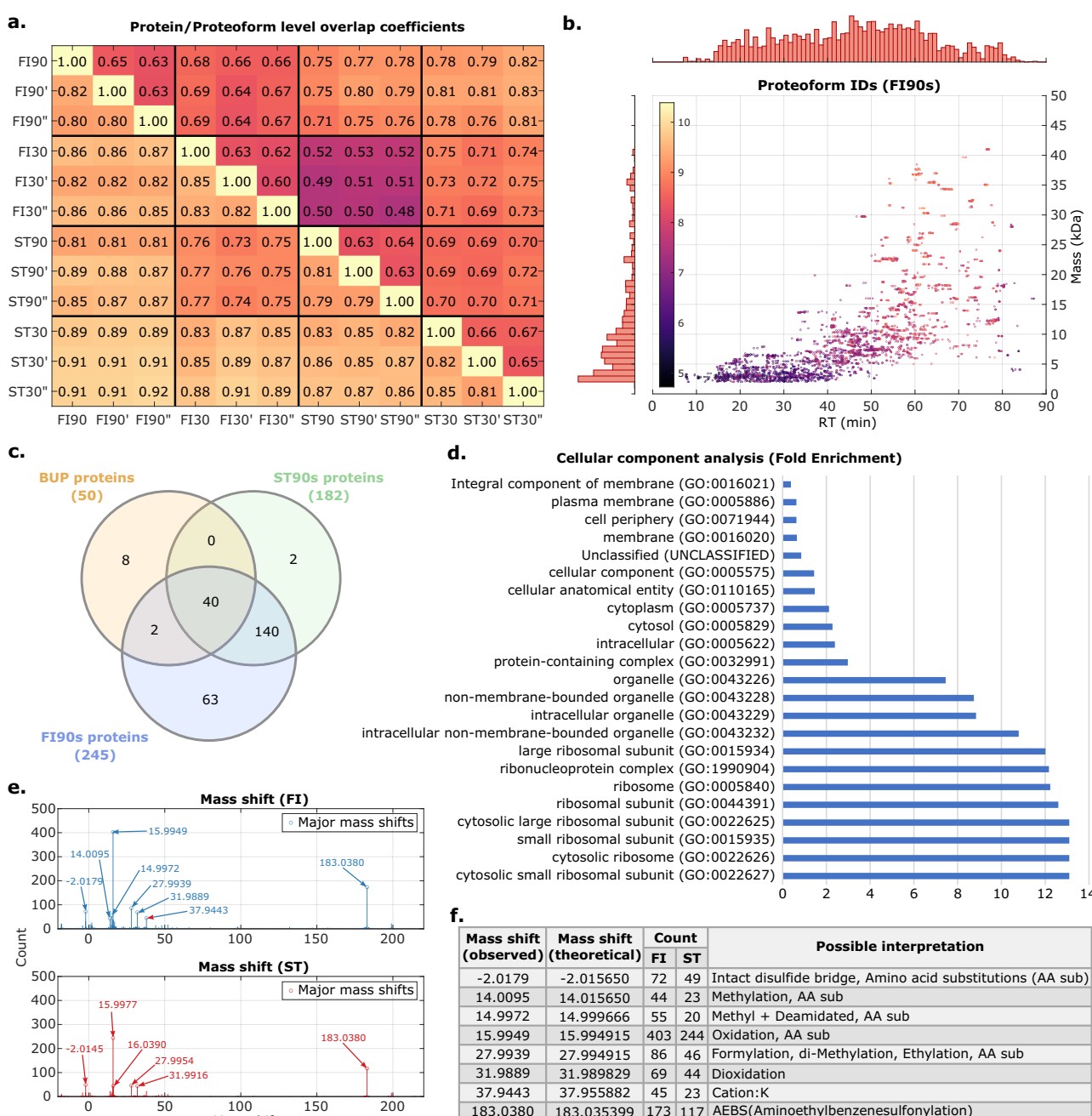

**Fig. 3 Analysis on proteoform IDs. a** Protein (lower diagonal) and proteoform (upper) level overlap coefficients over all datasets. FI90, FI90', and FI90" denote the triplicate datasets of FI90s, and the same notation applies for the others. FLASHIda datasets showed comparable overlap coefficients with the standard, even with significantly larger ID counts than the standard. **b** Scatter plot showing the proteoform IDs from FI90s datasets over retention time (RT) and mass with marginal histograms. Each ID is color coded by its intensity. Most proteoform IDs are present in the low mass region (<20 kDa) and RTs from 10 to 60 min. **c** Comparison between our TDP identified proteins with 50 high abundant BUP identified *E. coli* proteins (in Schmidt et al.[33]). From FI90s and ST90s, 245 and 182 proteins jointly detected in all replicates were collected, respectively. Out of 50 BUP proteins, 42 and 40 overlapped with the FI90s and ST90s proteins, respectively. Six of the eight undetected proteins were heavy proteins (>35 kDa; see Supplementary Data 11). We also found that the FI90s proteins contain most of the ST90s proteins (~99%). **d** Cellular component gene ontology (GO) analysis for the proteins identified from FI90s triplicates, using the Gene Ontology Resource[34, 35] (see Supplementary Data 12). The most abundant proteins are ribosomal proteins, as reported in Ishihama et al.[36]. Relatively rare membrane proteins were reported, indicating that analyzing intact membrane proteins via TDP is still a challenge. **e** Histograms of mass shifts (with a bin size of 0.05 Da) in proteoform IDs from all FLASHIda runs (FI; top panel) and standard (ST; bottom). The circled shifts are the ones that occur more than one tenth of the Oxidation shift. Each annotated shift value is the median of the shift values within its corresponding bin. The mass shifts were highly consistent for both acquisition methods. Source data are provided as a Source Data file. **f** Interpretations of major (circled) mass shifts found in **e** from Unimod[37]. The shift values in the first column are from FI datasets. The third and fourth columns show the numbers of found mass shifts in FI and ST datasets, respectively.

information even for families of proteoforms including low-abundance species.

We then focused on the modifications in the proteoform IDs in all FI datasets. TopPIC search allows for one modification per proteoform ID represented by a mass shift. Out of all proteoform IDs from FLASHIda runs, about 52% contained modifications. In the case of standard DDA runs, 43% were modified proteoforms. Figure 3e, f show the frequent mass shifts found in the proteoform IDs (in both FI and ST datasets) and their interpretations. For the interpretations, we referred to Unimod[37]. Even if we did not specify any candidate modifications (or mass shifts) except for N-term modifications, most of the mass shifts could be explained by well-known modifications such as methylation, oxidation, and amino acid substitution. A large mass shift of +183.04 Da was also observed, which corresponds to artifacts from using AEBSF (Pefabloc) as a serine protease inhibitor during sample preparation[38]. Figure 3e, f also show that the number of modified proteoform IDs in the FI datasets are almost twice those of the ST datasets (similar to unmodified proteoform IDs). Moreover, when the same frequency threshold (>10% the most frequent mass shift, which is oxidation) for the modification annotation was applied, FI datasets yielded exclusive annotations such as methylation and replacement of a proton by potassium.

The remaining 48% of all proteoform IDs from FI datasets contained no mass shift, but many were truncated species. N-terminal methionine removal was present in about 30% of all proteoform IDs. In several proteins with well-documented signal peptides, we observed signal-peptide cleavage at the reported cleavage site (e.g., in metal-binding protein ZinT (UniProtKB: P76344) and inhibitor of g-type lysozyme (UniProtKB: P76002)). In the case of nickel/cobalt homeostasis protein RcnB (UniProtKB: P64534), the starting position of all proteoform IDs was one residue after the reported site (determined by Edman sequencing[39]), suggesting possible further processing after signal peptide cleavage or presence of alternative start codons. In addition to them, many putative proteins with signal peptides have been identified in FI and ST datasets, including many FI exclusive ones. For instance, when we collected only unmodified proteoform IDs (to remove possible false positives) with the truncation of the N-terminal 16–30 amino acids, 50 and 37 proteoform IDs were found from FI90 and ST90 datasets, respectively. Out of them, 12 were FI90 dataset exclusive, containing ribose import binding protein RbsB (UniProtKB: P02925) with known signal peptide cleavage. In total, 600 unmodified and truncated proteoforms have been identified from the FI90 dataset while 375 have been identified from the ST90 dataset. Thus we can conclude that FLASHIda enables more sensitive detection of proteins and proteoforms not seen with standard acquisition in complex sample studies.

Supplementary Data 13 reports 419 proteoform IDs in FI datasets from top four selected proteins (two ribosomal proteins and two binding proteins; UniProtKB: P0ACF8, P0A7N1, P76344, P60438) with the four highest proteoform ID diversity. It also provides the manual validation results and possible interpretations for the modifications (see also Supplementary Figs. 23–27 for example annotations). In the two ribosomal proteins, common modifications were frequently observed such as oxidation and methylation. In addition, modifications like dioxidation, dimethylation, and unexpected amino acid substitutions were also reported. In two binding proteins, many proteoform IDs were reported to have large mass shifts (ranging from 200 to 400 Da), representing bound metal ions or nucleotides as well as chemical adducts.

To exclude chemical adducts and to further remove possible false modifications, we also ran TopPIC for FI90s and ST90s

datasets with eight important candidate modifications including acetylation, phosphorylation, and formylation (see Supplementary Table 1 and Supplementary Data 14 for information on the eight candidate modifications). After the search, the proteoform IDs containing mass shifts not explained by the candidate modifications were discarded (see Supplementary Data 15 for the result). In this stringent search, the proteoform ID count in FI90s again outnumbered that in ST90s by 62% (976 vs. 602 on average). Moreover, the proteoform IDs with lysine acetylation were exclusively found in FI90s datasets.

## Discussion

The potential of IDA is beginning to emerge gradually, as seen in the development of many IDA methods for various MS-based applications such as TDP, BUP, and metabolomics[12,26,40–43]. The data acquisition algorithm presented here shows a drastic performance boost in terms of sensitivity and throughput for complex TDP samples. Since it does not require any change in the experimental set up, but is rather a software extension module to the instrument software used for data acquisition, we believe that FLASHIda could be readily adopted for TDP studies to increase proteoform coverage at no additional cost.

Our algorithm currently focuses on the intelligent online selection of precursors, but it could obviously be improved to provide more fine control of the instrument, such as control of dissociation and ion injection duration tailored to the triggered precursor. For instance, from the uneven distribution of masses in Fig. 3b, one can find the need for different selection strategies for different proteoform ion mass or RT ranges. While only HCD was used for this study, other dissociation methods, such as electron-transfer dissociation (ETD), could be dynamically selected for larger molecules to obtain better fragmentation of proteoforms[44]. The overhead to switch between dissociation methods could be compensated for by an increased identification rate of large proteoforms. Also the ion injection time could be adjusted based on the number of available precursor masses. From the data acquired for this study, we found that many of the MS spectra contain less than four candidate precursors owing to either sparse proteoform ions or low precursor qualities (see Fig. 3b and Supplementary Fig. 12). In such cases, increasing the injection time for MS2 generation or selecting multiple precursors for the same proteoform mass (of distinct charges) as in Lu et al.[12] could lead to a higher number of proteoform IDs.

It is noteworthy that QScore in FLASHIda replaces the real-time identification in Autopilot, reducing the need for additional compute power. We demonstrated that QScore accurately measures how likely the selected precursor leads to successful identification in our tested datasets. However, the accuracy of QScore deteriorates with different MS datasets; for example, when ETD Orbitrap spectra from E. coli lysate (published in Dupre et al.[5]) were analyzed, QScore underestimated the actual identification rate by 30%. On the other hand, for the CID QTOF spectra from the pig heart sample (published in Brown et al.[45]), QScore overestimated the identification rate by 30%. After retraining of QScore weights, the estimation error returned to within 5% in both cases. While QScore is a quality measure rather than an identification rate estimator and thus estimation inaccuracy has only a minor effect on the performance of FLASHIda, this inaccuracy may still lead to suboptimal proteoform identification results. In addition, the QScore and TQScore thresholds (0.25 and 0.9, respectively) used in this study were determined rather empirically and manually; statistical methods to determine optimal thresholds could yield most proteoform IDs for different setups. Thus, while QScore alleviates the need for real-time identification, it may still require retraining of feature weights for

different instrumental setups. We plan to implement an automated training pipeline within FLASHDeconv in the near future.

We also note that the precursor mass determination error and coeluted precursors would be one of the major issues in comprehensive TDP analysis and part of our future work. Many studies discussed FDR control in TDP (e.g., LeDuc et al.[46]), but this issue remains rarely discussed. We showed that such precursors lead to false proteoform discovery not measured nor controlled by conventional target-decoy approaches (Supplementary Figs. 5, 6 and Supplementary Data 1). Therefore, the current version of FLASHIda focuses on obtaining precursors with low interference to minimize precursor errors as well as to avoid coeluted precursors (using precursor SNR). However, with respect to coeluted precursors, FLASHIda would allow acquisition of chimeric spectra from them if an appropriate precursor mass error estimation method were developed. Dedicated algorithms for the identification of chimeric spectra could then increase the identification rate further.

The current version of FLASHIda supports proteoform identification by TopPIC due to the absence of interface between FLASHDeconv (for spectral deconvolution) and other proteoform identification tools like ProSight PC. Also, as stated above, the interface between FLASHIda and other deconvolution tools such as Xtract[47] and TopFD could be implemented to improve identification performance. Thus, one of the future directions would be to develop such interfaces to easily deploy FLASHIda in various existing proteoform analysis pipelines.

While this study showcased the use of IDA for comprehensive TDP studies, different variations of the algorithm are currently under development for targeted proteoform analyses like global targeting[41], deep characterization, and even de novo sequencing. For such applications, information on the target such as sequence, homologous proteoforms, composition, and candidate modifications could be used as inputs to FLASHIda for better acquisition tailored to the needs of a specific study. Furthermore, as FLASHIda enables the selection of interference-free precursors, it could be used to enhance the quantification accuracy of isobaric labeling-based proteoform quantification[48]. We anticipate that advanced data acquisition methods (to be) developed within FLASHIda would facilitate exploration of proteoform heterogeneity via TDP.

## Methods

***E. coli* sample preparation**. *E. coli* protein sample (Biorad, USA, cat#1632110) was dissolved in 0.1% formic acid solution to a final concentration of 1 g/L. Proteins were desalted with a solid-phase extraction tip packed with Poros R2 packing material (Thermo Scientific, USA) before the analysis.

*LC-MS/MS*. A two-column setup was connected to EASY-nLC nano-HPLC (Thermo Scientific, San Jose, CA). Both the column (75 μm ID, 25 cm) and the precolumn (100 μm ID, 3 cm) were packed in-house using PLRP-S (1000 Å, 5 μm) packing material (Agilent, USA). Two different gradient lengths were used for both the standard and FLASHIda acquisition methods. Samples were recorded in triplicates for each condition, and 1 μg of the desalted sample was loaded for each run. Order of runs was randomized, and a column wash was included after each *E. coli* LC-MS/MS run to minimize the carryover. Mobile phase A consisted of 1% ACN 0.1% FA in water, and phase B of 95% ACN and 0.1% FA. In the 90 min runs, the concentration of buffer B was 5% for 5 min, 20% at 5 min, 55% at 70 min, 90% at 72 min, and 5% for the last 10 min. For the 30 min runs, it increased from 5% to 20% for the first two min, and was 55% at 19 min, 90% from 20 to 25 min, and 5% for the last 5 min.

Orbitrap Eclipse Tribrid mass spectrometer was used in the intact protein mode. Transfer tube temperature was set to 305 °C and 2.2 kV spray voltage was applied. Orbitrap resolution was set to 120,000 for MS1, and 60,000 for MS2 scans. Scan range was 500–2,000 and 400–2,000 Th for MS1 and MS2 scans respectively. HCD fragmentation was used with 29% normalized collision energy and 3 Th isolation window for ST datasets and dynamic isolation window size for FI datasets (see below). Maximum injection time was 50 ms (MS1) and 500 ms (MS2) and AGC target was 200% (8e5) (MS1) and 1000% (5e5) (MS2). Only the precursors with charge states 4–50 were selected for fragmentation in both standard and

FLASHIda runs. For standard runs, dynamic exclusion was applied with a duration of 30 sec and m/z tolerance of 1.5.

*FLASHIda software structure*. FLASHIda is a console application written in C# (interaction with mass spectrometer) and C++ (deconvolution by FLASHDeconv, precursor selection, quality based exclusion, etc). It uses the instrument API (iAPI) provided by Thermo Scientific to communicate with the mass spectrometer. FLASHIda only can be used in iAPI compatible tribrid mass spectrometer series with instrument software version 3.4. The access to iAPI is available upon request from Thermo Scientific and is subject to a separate license. The parameters of acquisition (MS1 and MS2 scan parameters, and FLASHDeconv parameters) are controlled by a method file in XML format. Survey MS1 scans are received from the instrument through the iAPI in a vendor-specific format; briefly, the spectrum consists of an array of peak centroids and additional spectral metadata such as retention time, polarity, analyzer type, etc. Next, the spectra are converted into a format compatible with FLASHDeconv (separate m/z and intensity arrays) and sent to FLASHDeconv part via platform invocation (PInvoke). Upon receiving the spectrum, FLASHDeconv processes the spectrum and selects *N* (user controlled parameter) high quality precursors. The isolation window covers the complete precursor envelope range with 0.6 Th margin from both sides. On average, the isolation window size from FI datasets was 3.1 Th.

The list of precursors (or isolation windows) returned by FLASHDeconv is used to create fragmentation scans in instrument specific format, which are added to a FIFO queue to be sent to the instrument. A survey MS1 scan as well as AGC ion trap scan is added to the queue after each set of fragmentation scans.

*QScore calculation via logistic regression*. After the deconvolution, firstly per charge QScore, denoted as $\mathbf{Q}(m, z)$, is calculated for each precursor of (monoisotopic) mass $m$ and charge $z$ using a logistic regression. $\mathbf{Q}(m, z)$ is the probability that the MS2 spectrum from the charge $z$ precursor of mass $m$ is identified. The QScore of mass $m$, denoted as $\mathbf{Q}(m)$, is simply given by the maximum of $\mathbf{Q}(m, z)$ for all charges $z$. And the charge $z$ such that $\mathbf{Q}(m, z)=\mathbf{Q}(m)$ becomes the charge of the precursor.

Denote the mass of the proton by $c$, and the mass difference between $^{13}$C and $^{12}$C by $\delta$. Given a mass $m$ and charge $z$, if a peak has m/z of $((m + n\delta)/z)+c$ for positive MS mode or m/z of $((m + n\delta)/z)-c$ for negative mode within (user defined) tolerance, the peak is said to correspond to the *n*-th isotope (of mass $m$ and charge $z$). Denote an isotope envelope of mass $m$ and $z$ as a vector $\mathbf{E}(m, z)=(i_1, i_2, ..., i_n)$, where $i_n$ denotes the summed intensity of the peaks corresponding to the *n*-th isotope. Suppose the charge range of this envelope is from $v$ to $w$. Then the aggregated envelope $\mathbf{E}(m)$ is a vector given by adding all vectors $\mathbf{E}(m, z)$ for $z = v,...,w$ element wise. $\mathbf{Q}(m, z)$ is furnished by a logistic regression with six features, two extracted from the charge $z$ envelope, $\mathbf{E}(m, z)$, and four from the aggregated envelope $\mathbf{E}(m)$. The features are given by

1. Cosine similarity between the isotope pattern of $\mathbf{E}(m, z)$ and the theoretical one: charge_cos
2. Precursor SNR (signal to noise ratio) or SNR of $\mathbf{E}(m, z)$: precursor_SNR
3. Cosine similarity between the isotope pattern of $\mathbf{E}(m)$ and the theoretical one: mass_cos
4. Mass-level SNR or SNR of $\mathbf{E}(m)$: mass_SNR
5. Charge distribution score representing how evenly peak intensities are distributed along different charges: charge_dist
6. Average mass error of $\mathbf{E}(m)$ in PPM unit: mass_error

The theoretical isotope patterns are obtained by using averagine model[49]. The cosine similarity was calculated between observed and the theoretical patterns, as described in Jeong et al.[29]. The SNRs consider the peak locations as well as the shape of the isotope pattern for better separation of signal and noise components (see below). The charge distribution score calculation and average mass error calculation are described below as well. For cosine similarities and charge distribution scores $x$, we use $log_2(1 + x)$ as feature values. For SNRs $y$, we use $log_2(1 + y/(1 + y))$ as feature values. The mass error values are used as is.

From the training of our logistic regression model, the weights for the QScore are determined by 0.4074, −1.5867, −22.1376, 0.4664, −0.4767, 0.541, and 20.248, for charge_cos, precursor_SNR, mass_cos, mass_SNR, charge_dist, mass_error, and intercept, respectively. From the weights, mass_cos is shown to be the most important feature for quality measure.

To determine informative features, we first included only three features, charge_cos, charge_dist, and mass_cos as they are directly available from FLASHDeconv. In 10-fold cross validation, the trained model resulted in ROC (Receiver operating characteristic) area of 0. 775. However, we found that only with these features, coeluted precursors or harmonic masses are not penalized enough. Thus we added two more SNR related features: precursor_SNR and mass_SNR. The resulting ROC area increased to 0.79. Lastly, we added mass_error achieving the ROC area of 0.80. The addition of peak or mass intensity features almost did not improve the classification performance (ROC area of 0.80). But precursor mass and charge related features improved the classification (ROC area of 0.87). This boost, however, introduced a strong bias towards smaller proteoform masses (<20 kDa) resulting in drastically reduced number of heavy proteoform IDs. Moreover, the optimal weight for the precursor mass or charge feature may be strongly dependent on instrumental

settings, e.g., resolution, making the model less robust than without it. Thus we finalized the six features listed as above.

To train the logistic regression model, six *E. coli* single run datasets were analyzed (deconvolved by FLASHDeconv and identified by TopPIC at 1% spectrum and proteoform level FDR). The identified 13,205 precursors and unidentified 18,261 precursors were used as true and false classes, respectively. The training was done by Weka software (version 3.9.4)[50] with default parameters for logistic regression. The input files to Weka software are found in Supplementary Data 16.

*SNR estimation.* We first describe how the precursor SNR of the charge envelope $E(m, z)$ is measured. Given a charge envelope $E(m, z)$, denote the start m/z and the end m/z of $E(m, z)$ as $t$ and $s$, respectively. Out of all peaks between $t$ and $s$, the peaks not corresponding to any isotope are noisy peaks and have only a noise component. The sum of squared intensities, or power, of these noisy peaks is written as $N_1$. The peaks corresponding to isotopes mainly have a signal component of $E(m, z)$, but also have a noise component from signal distortion, coelution, or simple thermal additive noise. This noise power (denoted by $N_2$) and signal power (denoted by $S$) within such peaks can be measured by solving the least square problem described below.

Denote the intensities of the theoretical isotopic distribution (from the averagine model[49]) of mass $m$ by a vector $V = (v_1, v_2, ..., v_n)$, where $v_i$ denotes $i$-th isotope intensity. Denote $E(m, z)$ as $E$ for simplicity. We want to know how much of the $V$ component exists in $E$. To do so, we find a weight $w$ such that $||V - wE||^2$ is minimized. This is a typical least square problem and the solution for $w$ is given by

$$w = (E^T E)^{-1} E^T V \qquad (1)$$

or equivalently,

$$w = \cos\theta \|V\| / \|E\| \qquad (2)$$

where $\cos\theta$ is the cosine similarity between $E$ and $V$, which is already available from FLASHDeconv. Once the weight $w$ is calculated, the signal component power in $V$ is given by $S = ||wE||^2 = \cos^2\theta \|V\|^2$. The noise component power in $V$ is given by $N_2 = \sin^2\theta \|V\|^2$.

With the calculated signal power $S$ and total noise power $(N_1 + N_2)$, the SNR is furnished by

$$SNR = S/(N_1 + N_2) = \cos^2\theta \|V\|^2 / (N_1 + \sin^2\theta \|V\|^2) \qquad (3)$$

Note that this definition considers not only the noisy peaks outside the isotope locations but also those coexisting with the peaks corresponding to the isotopes.

The mass SNR for $E(m)$ is done in the same way, except that the peaks of all charges are used in the place of charge $z$ peaks for $E(m, z)$. The calculation of SNR is implemented in FLASHDeconv without loss of computational efficiency.

*Charge distribution score calculation.* The charge distribution score is calculated as follows. Given a mass $m$, let $i_z$ denote the summed intensity of the charge $z$ peaks corresponding to any isotopes of the mass $m$. Further denote the minimum charge by $w$ and the maximum charge $W$. Then for the mass $m$, we have a vector $(i_w, ..., i_W)$ that represents per charge intensity of $m$. Let $x$ be the charge with the maximum per charge intensity, that is, the charge such that $i_x \geq i_z$ for all $z = w, ..., W$. Then we define the left penalty $l_z$ for $z = w + 1, ..., x$ by $l_z = \max(0, i_{z-1} - i_z)$ and the right penalty $r_z$ for $z = x, ..., W-1$ by $r_z = \max(0, i_{z+1} - i_z)$. The total penalty $P$ is given by the summation of left and right penalties. Let $I$ be the summation of all $i_z$ values. The charge distribution score is given by $1 - P/I$.

*Average mass error calculation.* Given the aggregated envelope $E(m)$, the peaks corresponding to isotopes of $E(m)$ may have different isotope indices. For a charge $z$ peak of isotope index $n$, its monoisotopic mass $p$ is calculated by $p = z(t - c) - n\delta$ for positive MS or $p = z(t + c) - n\delta$ for negative MS, where $t$ is the m/z of the peak, $c$ denotes the mass of proton, and $\delta$ the mass difference between $^{13}C$ and $^{12}C$. Then its PPM error is given by $10^6 |(m - p)/m|$. The average mass error is given by the average value of these PPM errors over all the isotope corresponding peaks.

*TQScore (Total QScore) calculation.* For a mass $m$, assume that it has been triggered $n$ times with QScores $q_1, ..., q_n$. TQScore of $m$ is calculated by $1-(1 - q_1)(1 - q_2)...(1 - q_n)$. This calculation assumes mutual independence among MS2 spectra of precursor mass $m$ in terms of their identification rates. This is not rigorous in particular when multiple MS2 spectra are acquired from the precursors of the same charge state as they are expected to be highly similar to each other. However, since the primary goal of TQScore is not to estimate the identification rate accurately but to measure the (rough) quality of the mass for acquisition, we took this simple definition. To improve the estimation accuracy, methods taking the correlation between MS2 spectra into account should be applied, such as the one using Bayesian network.

*Target-decoy based FDR estimation of MS2 spectra of incorrect precursor masses.* To test the effect of incorrect precursor masses on FDR estimation via target-decoy

approaches, we took the deconvolved spectra from FI90 dataset (by FLASHDeconv) and generated two sets of deconvolved spectra with false precursor masses. The first set is generated by dividing all the precursor masses by two, simulating low harmonic artifacts (as they are the most common precursor errors from our observation). The second set was generated by adding arbitrary values ranging from $-10$ to $-1$ or from 1 to 10 to the precursor masses. Then these sets were identified using TopPIC at 1% spectrum and proteoform level FDR from the target-decoy approach. The results are shown in Supplementary Fig. 2, 3 and Supplementary Data 1.

*FLASHDeconv commands for analysis of FI and ST datasets*
For FI90 dataset:
```
FLASHDeconv -in [input mzML] -out [feature deconvolution out
tsv] -in_log [FLASHIda log file] -out_topFD [ms1 deconvolved
spectrum msalign] [ms2 deconvolved spectrum msalign] -out_
topFD_feature [ms1 feature] [ms2 feature] -min_precursor_
snr 1.0 -Algorithm:min_charge 4 -Algorithm:max_charge 50
-Algorithm:rt_window 180 -Algorithm:min_mass 500 -Algo
rithm:max_mass 50000
```

For FI30 datasets:
```
FLASHDeconv -in [input mzML] -out [feature deconvolution out
tsv] -in_log [FLASHIda log file] -out_topFD [ms1 deconvolved
spectrum msalign] [ms2 deconvolved spectrum msalign] -out_
topFD_feature [ms1 feature] [ms2 feature] -min_precursor_
snr 1.0 -Algorithm:min_charge 4 -Algorithm:max_charge 50
-Algorithm:rt_window 60 -Algorithm:min_mass 500 -Algo
rithm:max_mass 50000
```

For ST90 datasets:
```
FLASHDeconv -in [input mzML] -out [feature deconvolution out
tsv] -out_topFD [ms1 deconvolved spectrum msalign] [ms2
deconvolved spectrum msalign] -out_topFD_feature [ms1 fea
ture] [ms2 feature] -min_precursor_snr 1.0 -Algorithm:min_
charge 4 -Algorithm:max_charge 50 -Algorithm:rt_window 180
-Algorithm:min_mass 500 -Algorithm:max_mass 50000
```

For ST30 datasets:
```
FLASHDeconv -in [input mzML] -out [feature deconvolution out
tsv] -out_topFD [ms1 deconvolved spectrum msalign] [ms2
deconvolved spectrum msalign] -out_topFD_feature [ms1 fea
ture] [ms2 feature] -min_precursor_snr 1.0 -Algorithm:min_
charge 4 -Algorithm:max_charge 50 -Algorithm:rt_window 60
-Algorithm:min_mass 500 -Algorithm:max_mass 50000
```

The difference between FI and ST datasets is that in FI datasets, FLASHIda log file name is specified with the `in_log` option. The only difference between 90 and 30 min runs is the `rt_window` parameter value that decides the internal RT window size for FLASHDeconv. FLASHDeconv considers the spectra within this window for sensitive mass deconvolution (see Jeong et al.[29]). The generated msalign and feature files are used as the inputs to TopPIC.

*TopPIC commands for analysis of FI and ST datasets.* For all datasets, we used the same command

```
toppic.exe -d -t FDR -T FDR -u 16 [fasta] [ms2 msalign file
name]
```

For the search with candidate modifications (for Supplementary Table 1 and Supplementary Data 14), we added the `-i` option as in:

```
toppic.exe -d -t FDR -T FDR -u 16 [fasta] [ms2 msalign file
name] -i [modification text file name]
```

The input modification text file for this search is given in Supplementary Data 14.

**Reporting summary**. Further information on research design is available in the Nature Research Reporting Summary linked to this article.

## Data availability
The raw and converted mzML files for FI and ST triplicate datasets as well as the identification results and MS1 signal screenshots (related to Supplementary Figs. 2–4,

18–20, and 24–27) have been uploaded to MassIVE under accession number MSV000087484 https://doi.org/10.25345/C5FJ9G. Source data are provided with this paper.

## Code availability

The C# source code of FLASHIda is published under a BSD three-clause license and is available at https://OpenMS.de/FLASHIda with the link to Github repository (https://github.com/caetera/FlashIda) containing detailed information on building and using FLASHIda. FLASHDeconv is implemented in C++ as a part of OpenMS[51] and available as platform-independent open-source software under a BSD three-clause license at https://OpenMS.de/FLASHDeconv (Github repository: https://github.com/JeeH-K/OpenMS/tree/feature/FLASHDeconv).

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

## Acknowledgements

The authors acknowledge Jesse Canterbury and Shannon Eliuk from Thermo Fisher for their help with iAPI use. K.J. is thankful to Xiaowen Liu for insightful advice on the data analysis. Proteomics and mass spectrometry research at SDU are supported by generous grants to the VILLUM Center for Bioanalytical Sciences (VILLUM Foundation grant no. 7292 to O.N.J.) and PRO-MS: Danish National Mass Spectrometry Platform for Functional Proteomics (grant no. 5072-00007B to O.N.J.). K.J., J.K., M.B., O.N.J., and O.K. acknowledge funding from the Horizon 2020 Marie Sklodowska-Curie Action ITN 2017 of the European Commission (grant 765502-A4B). K.J. and O.K. acknowledge EPIC-XS (project number 823839), funded by the Horizon 2020 program of the European Union.

## Author contributions

O.K. conceived the idea of the real-time deconvolution-based intelligent acquisition control for TDP datasets. K.J. and V.G. developed and implemented FLASHIda algorithm. M.B. and O.N.J designed the experiment and performed sample preparation and data acquisition. All authors analyzed the datasets. O.K. and O.N.J led the project and provided resources. K.J. and J.K. wrote the manuscript with input from all authors. All authors commented on and approved the paper.

## Funding

## Competing interests

The authors declare no competing interests.
