## [Peer Review File · Nature Communications]

REVIEWER COMMENTS

Reviewer #1 (Remarks to the Author):

In this manuscript, Jeong et al describe the development of an online intelligent data acquisition platform that results in an increase in the number of proteoform identifications. Acquiring high quality MS2 spectra of a variety of proteoforms is a difficult challenge in LC-MS workflows without freely available software solutions. This manuscript attempts to address this challenge and importantly provides this software to be freely available for other labs to implement; however, I think some further discussion is needed and some of the claims need more analysis:

1) In the introduction line ~73, I think this sentence needs more clarification. For example is the 3-5% number from AutoPilot or MetaDrive? And did both of these manuscripts attribute limitations to deconvolution speed, specifically? Since AutoPilot is one of the main previous works with a similar idea to this manuscript, I think a comment would be interesting on their approach (online database searching for precursor selection) vs. this approach (determining a Q-score from the MS1-level data to determine which precursors to select).

2) How general is the trained Q-score prediction machine learning model? Do the authors suspect this score will need to be retrained for different LC columns/parameters or instrument parameters? Some discussion or analysis would be helpful to show this is a general approach that can be implemented on other labs' instrument setups. For example, the Q-score calculation could be applied to other published datasets (from another lab even) to determine how well it predicts the identification of the spectra (without needing to collect additional data).

3) Although identification counts are important, further analysis would be helpful to show that using FLASHIda improves the proteoform identifications. How does MS2 spectral quality track with Qscore and how did trends in MS2 quality improve with FLASHIda vs. standard analysis? Are the identifications higher quality and more confident with FLASHIda, such as better sequence coverage or other MS2 quality metric? Were PTMs better localized?

4) The fact that the improvement was only 5-10% when using TopFD deconvolution seems to be an important point that did not have much discussion and maybe requires further exploration... If ~60% deconvoluted masses overlapped with a SNR of 1, what explains the low ID improvement with FLASHIda when using TopFD deconvolution? Were all of the additional IDs in the TopFD analysis part of the 40% deconvoluted proteoforms that did not overlap with the SNR threshold?

5) In line 203, the authors mention a boost being "recovered to 50-65%" ... how do these results analyzed here differ from the previous paragraph that mentioned a doubling of IDs and also used TopFD and FlashDeconv as the deconvolution algorithm? Some clarification here is needed.

6) I think the paragraphs describing the proteoform identifications under "Analysis of proteoform and protein coverage" section would be more relevant to the paper and interesting if the discussion

really highlighted how FLASHida enabled these results... for example, a few paragraphs describe the signal peptides, truncated proteoforms, and modified proteoforms identified in FLASHida, but it would be interesting to know if these were also identified in the standard runs? If they were identified in standard runs, this discussion is not very relevant to the paper; if these proteoforms were not identified in the standard run, this fact should be much more highlighted as an interesting result and strength of the platform to reveal additional proteoform biology.

7) In general, I think the paper needs better discussion and analysis of additional proteoform IDs (in the doubling ID count for standard vs. FLASHida) - for example, were these proteoforms selected for fragmentation in the standard runs at all? Were they observed in the standard runs but not selected? Were they fragmented in the standard runs but only once due to the simple mass exclusion list used? Additionally, finding a few examples that validate the claim that FLASHida enabled identification of proteoforms that would not be identified by standard data acquisition would be helpful, such as for a few of the biologically interesting proteoforms discussed in comment #6.

Reviewer #2 (Remarks to the Author):

Overall, I find this work to be very well done and it is a welcomed addition to the field of top down proteomics. The incorporation of online FLASHDecon and the Qscore approach to precursor selection is novel and shows a marked improvement in both protein and proteoform identification. The C# source code is well written and well documented. I did feel, however, that the discussion around Figure 3B and 3D was superfluous and not necessarily supportive of FLASHida. I am also concerned that the noteworthiness of this work is limited in light of past works of Durbin, Fornelli, and Lu.

1. (Line 63): Is mass-based exclusion not common on modern instruments? I was under the impression that this was now standard during top down acquisition.
2. (Line 67): I feel that the range of 5-15 Th for an isolation is a bit dated; recently released instrumentation commonly use a quadrupole to isolate in the 1.2 - 3 Th range. Lubeckyj et al. (18) mentioned a window of 4Th and Toby et al. (5) mentions 15Th only as 'ultra-wide isolation'. Isolation width is very important in proteoform discovery, but I don't think this approach improves on the current state of the art in this way. Line 462 mentions that you used a width of 3 Th.
3. (Line 74): Two crucial citations were missing that call into question some assertions made in this section. Given the age of Durbin et al., a follow up "Autopilot" search turned up 2 follow up papers: "Quantitation and Identification of Thousands of Human Proteoforms below 30 kDa", Durbin et al.

2016 and "Advancing Top-down Analysis of the Human Proteome Using a Benchtop Quadrupole-Orbitrap Mass Spectrometer", Fornelli et al. 2017.

4. There were a couple word choices that I would change. What does 'divertize' mean (lines 71 and 675)? Please revise the word 'instant' to be more quantitative (line 93).

5. I really liked how you structured your logistic regression (starting on line 98) and how thoughtfully you laid out the six features (Methods).

6. (Line 104): Could you comment more on the preliminary studies you conducted to determine features?

7. It was mentioned that the TQScore threshold was 0.9 (lines 130, 147) and the QScore threshold was 0.25 (line 148). How were these arrived at?

8. (Line 131): I'm not sure what '(of proteoform)' means? Perhaps a typo?

9. (Line 147): 400 Da seems very low for a mass cutoff. Given your scan range was 500-2000 Th (line 461) and your lower charge limit was 4 (line 464), wouldn't 2000 Da be a more appropriate limit?

10. (Line 167): It is notoriously difficult to download a specific version of a UniProt database after the fact. Please consider including an archived fasta file in your supplemental data.

11. I appreciate that you were careful in how you reported protein and proteoform counts (around line 169). Choosing to compare the average across the technical replicates, and not some naïve "total" count, preserves control of the FDR.

12. (Figure 2A): Why does the ST90s bar on # proteoform IDs not match Supplemental Figure 13c?

13. (Figure 2B, right): I noticed that the ST experiment (top) targets a small population of higher intensity precursors (i.e., the histogram on the ST chart goes farther to the right than the histogram for the FI chart). Can you speculate as to why this is the case?

14. I don't think the claim of the subsection title "IDA improves reproducibility of proteoform level identification" on line 206 is fully verified. While I fully accept that you found more proteoforms, on line 212, the reproducibility comparison is described only as "comparable". Please make the improvement in reproducibility clearer or tone down to the claim of the subsection.

15. In the section "Analysis of proteoform and protein coverage" starting on line 221, I feel that a comparison of FI and ST should be included. For example, did the ST90 proteoforms over 30kDa outperform the McCool study? The dynamic range of proteoform intensities for FI90 is 5-6 orders of magnitude; how does this compare to ST90?

16. In the section discussing the GO analysis (lines 239-61), I feel that comparisons between FI and ST are needed. As it stands, this section only serves to validate that your workflow matches a previous BUP study and does little to bolster FLASHIda vs. 'standard' top down.

17. (Line 480): Is it correct to say that FLASHIda uses an isolation window of 1.2 Th ($2 * 0.6$ Th) and the standard workflow uses an isolation of 3 Th (line 462)? If so, why the difference? My concern is

that this difference might allow FI to pluck out specific proteoforms where ST would have found co-isolating chimeras (and been filtered).

18. (Line 483): Is this saying that you have 2 MS1 scans between each fragmentation scan? My understanding from the rest of the paper is that you have only a single MS1 scan (e.g., Figure 1A).

19. (Line 555): It seems that the extreme charges, w and W , are left out of the penalty calculation. Is this correct? If so, why?

20. In the C# source code, there is mention of a "magic scan identifier" of 41. What is this and why is it needed?

21. Are you planning to open source the code for the Qscore? I didn't see it in the C#.

22. As noted in the summary, I wanted to reiterate that the C# source code and documentation is well done and goes above and beyond what is typical. I enjoyed looking through it and plan to investigate the DataFlow library for some of my projects!

23. (Supplemental Figure 12): The caption says 2kDa, I believe it should be 20kDa.

Reviewer #3 (Remarks to the Author):

Development of the smarter data acquisition methods in mass spectrometry-based proteomics is one of the most important issues,

particularly for top-down proteomics. This is mainly due to the diversified MS1 signals of different charge states from the same proteoform. In this manuscript Kyowon et al. present an improved IDA algorithm, FLASHida, to resolve the problem of low coverage of proteoforms. FLASHida highly relies on FLASHDeconv, which is a fast algorithm on spectral deconvolution developed by the authors' lab. Much of the studies in this new manuscript went to algorithmic development to assess the quality of the masses (protein species in MS1) and then selected top-N

high-quality masses to trigger MS2 fragmentation according to their T-Qscores. Of all the experiments results presented, the most impressive one is the count of the identified proteoforms doubled by FLASHida when compared it to the standard DDA method.

In one word, the authors have presented an effective IDA algorithm and significantly improved the identification rate of top-down MS/MS. In order to improve the submitted manuscript further, the following suggestions to include more informative and convincing data results are listed.

Major:

1. In figure 2a, the number of precursors identified in FI90s is about 6000. The number of proteoforms identified in the same data set is about 1500. Therefore, each proteoform is identified four times on the average. Is this right? If it is right, why for the same proteoform, it is identified repeatedly? So, the comparison with DDA on the number of repeated identified proteoforms is suggested to be analyzed. That means for the E. coli data sets, on average, how many times MS/MS identified for each proteoform under FI90s and ST90s?

2. As we know, for most highly abundant proteoform, if it is not co-eluted and also with much less interference from other proteoforms, its experimental isotopic envelope will be well fitted to the computational one. However, in highly complex samples, their top-down data will not be so ideal. The co-eluted, the interference, and the low abundant skewed distribution will be everywhere. How to deal with those proportion more effectively?

For example, if there are two different proteoforms in the same isolation window, the peaks not included in one isotopic clusters are not necessarily noisy peak, but possibly from another proteoform. This will bias the SNR estimation if we ignore any one proteoform. Furthermore, how to assess their quality by Q-score?

This phenomenon is not atypical in typical top-down data sets.

3. For those high-quality acquired masses with MS/MS, there are still one quarter (25%) unidentified? what are the main reasons? Perhaps the one unknown PTM in TopPIC setting is the reason to lose those proteoforms with multiple modifications. Another is the FDR cut-off. So a simple analysis for all those 25%

unidentified will show some clues to improve the ID rate further. Searching those reasons is not a simple task because we do not the answer for some. However, we can also know some, such as listed above.

4. Most proteoforms are modified by PTMs as shown in Figures 3e and 3f. In figure 3f, please replace the integer mass shift by the accurate mass with at least four or six digits because some modification masses are the same when only use integer mass. What is the frequency of all those listed PTMs found according to the

ID results? Please give another column to show their appearance frequency or counts. For figure 3e, please annotate the accurate mass on those peaks.

5. In top-down data analysis by average model, the 'One-dalton' Shift is a troublesome question. This means there will be one-dalton or two-dalton mass shift between the prediction by average model and the accurate monoisotopic mass. This is validated by the results after identification of their MS/MS. So in the E coli data sets of F190s, please statistically analyze the one-dalton shift among all those precursors identified, such as what is those percentages, etc.

Minor

1. Line 40, the reference of 1 and 2 for proteoform is not the original paper. Please instead use the following original citation:

Proteoform: a single term describing protein complexity

Lloyd M Smith, Neil L Kelleher & The Consortium for Top Down Proteomics Nature Methods volume 10, pages186–187 (2013)

2.L59, 'inadequate'?, perhaps this sentence can be restated as, these criteria are suboptimal for high-quality proteoform ion selection.

3.L276, the alternative start codon can also be a reason?

4. Figure 2a, the right # precursors, the black, gray legend is not obvious to the bars below, which one is which one?

5. L563, for the negative ion mode, most precursors are deprotonated, so it is not electron. It is one proton lost for negative one charge.

REVIEWER COMMENTS

First of all, we are very grateful for all reviewers' insightful and extremely constructive comments, which helped us to improve the contents and the structure of the manuscript. We carefully addressed all the comments issued by all reviewers and corrected and supplemented the manuscript accordingly. In particular, additional comparative analysis between FLASHIda and standard DDA was added in different aspects including acquisition statistics, quality of acquired fragment spectra, and identified proteoforms (in particular FLASHIda exclusive identifications). All responses and corrections made in the manuscript are highlighted in blue.

Reviewer #1

In this manuscript, Jeong et al describe the development of an online intelligent data acquisition platform that results in an increase in the number of proteoform identifications. Acquiring high quality MS2 spectra of a variety of proteoforms is a difficult challenge in LC-MS workflows without freely available software solutions. This manuscript attempts to address this challenge and importantly provides this software to be freely available for other labs to implement; however, I think some further discussion is needed and some of the claims need more analysis:

1) In the introduction line ~73, I think this sentence needs more clarification. For example is the 3-5% number from AutoPilot or MetaDrive? And did both of these manuscripts attribute limitations to deconvolution speed, specifically?

The 3-5% improvements are from both tools. In the case of AutoPilot, the AutoPilot acquisition was compared against the standard acquisition with a low mass range GELFrEE fractionation of PAO1. The numbers of proteins were 234 for AutoPilot and 230 for the standard acquisition. The authors specified that this was due to narrow search space for AutoPilot real-time identification and clearly stated that "This search strategy was chosen to sufficiently limit the search space to enable searches to be performed on the same desktop computer that controls acquisition and finish within the cycle time of the instrument." We agree with the reviewer that this is not only about deconvolution speed, though.

In the case of MetaDrive, the authors were able to increase the proteoform identification rate by 10% by their acquisition method in yeast samples. But the increase in the proteoform ID count was limited due to the long deconvolution processing time, as the authors clearly stated that "the computation time for the real-time charge deconvolution algorithm is a source of delay in the decision-making process."

We rephrased the sentences on page 4 as follows:

“Both methods used real-time mass deconvolution (i.e., real-time determination of intact proteoform masses) to disentangle complex signal structure of proteoforms. Autopilot also employed real-time database search for better characterization of proteoforms as well as for less redundant precursor mass selection, while MetaDrive focused on improving the quality of the fragment spectrum by dissociating multiple precursor ions of the same proteoform. Both approaches clearly demonstrated the potential of IDA for TDP studies and, at the same time, were limited because of the need for faster processing times that could fit within the cycle time of the MS instrument.”

Since AutoPilot is one of the main previous works with a similar idea to this manuscript, I think a comment would be interesting on their approach (online database searching for precursor selection) vs. this approach (determining a Q-score from the MS1-level data to determine which precursors to select).

We thank the reviewer for this great comment. To address this comment and the very next one, we added the following paragraph in Discussion section (page 17-18):

*“It is noteworthy that QScore in FLASHIda replaces the real-time identification in Autopilot, reducing the need for additional compute power. We demonstrated that QScore accurately measures how likely the selected precursor leads to successful identification in our tested datasets. However, the accuracy of QScore deteriorates with different MS datasets; for example, when ETD Orbitrap spectra from *E. coli* lysate (published in Dupre et al.) were analyzed, QScore underestimated the actual identification rate by 30%. On the other hand, for the CID QTOF spectra from the pig heart sample (published in Brown et al.), QScore overestimated the identification rate by 30%. After retraining of QScore weights, the estimation error returned to within 5% in both cases. While QScore is a quality measure rather than an identification rate estimator and thus estimation inaccuracy has only a minor effect on the performance of FLASHIda, this inaccuracy may still lead to suboptimal proteoform identification results. In addition, the QScore and TQScore thresholds (0.25 and 0.9, respectively) used in this study were determined rather empirically and manually; statistical methods to determine optimal thresholds could yield most proteoform IDs for different setups. Thus, while QScore alleviates the need for real-time identification, it may still require retraining of feature weights for different instrumental setups. We plan to implement an automated training pipeline within FLASHDeconv in the near future.”*

2) How general is the trained Q-score prediction machine learning model? Do the authors suspect this score will need to be retrained for different LC columns/parameters or instrument parameters? Some discussion or analysis would be helpful to show this is a general approach that can be implemented on

other labs' instrument setups. For example, the Q-score calculation could be applied to other published datasets (from another lab even) to determine how well it predicts the identification of the spectra (without needing to collect additional data).

This is a very insightful comment. As we addressed above, we performed QScore calculation for different datasets: one from ETD Orbitrap and the other from CID QTOF runs. And indeed we observed over and underestimation of QScore. We discussed this point and specified the need for retraining of QScore for different instrumental setups in the Discussion section, as stated above.

3) Although identification counts are important, further analysis would be helpful to show that using FLASHIda improves the proteoform identifications. How does MS2 spectral quality track with Qscore and how did trends in MS2 quality improve with FLASHIda vs. standard analysis? Are the identifications higher quality and more confident with FLASHIda, such as better sequence coverage or other MS2 quality metric? Were PTMs better localized?

As shown in Fig. 2b left panels, the identification rate is proportional to QScore, which indicates that QScore and the quality of MS2 are directly related. However, when we plot $-\log_{10}(\text{evalue})$ reported by TopPIC (which can be considered a measure of MS2 spectrum quality) against QScores, we did not observe a strong correlation; the Pearson correlation coefficient was only about 0.03 for both FI and ST datasets. The likely explanation is that the identified MS2 spectra already exceed a rather high quality threshold.

Likewise, when we simply take two sets of all identified MS2 spectra (one from FI and the other from ST datasets), their quality differences (in terms of e-value, the number of annotated fragments, and the PTM localization ambiguity) were negligible. Next, for each FI and ST dataset, we collected 1,171 MS2 spectra from the jointly identified proteoforms (found in *proteoform.tsv output files by TopPIC). For these MS2 spectra, the average value of $-\log_{10}(\text{evalue})$ was higher for FI datasets than ST datasets (11.34 vs. 11.05). Moreover, the PTM localization ambiguity was lower for FI than ST (17.3 vs. 18.5). We added the following paragraph on page 10:

*“We then investigated the quality of MS2 spectra that are commonly identified by FLASHIda and standard DDA. To this end, from FI and ST datasets, we collected 1,171 jointly identified proteoform IDs. Then for each dataset, we took 1,171 MS2 spectra representing the collected proteoform IDs (found in *proteoform.tsv files reported by TopPIC). For these MS2 spectra, the average value of $-\log_{10}(\text{evalue})$ was higher for those from FI than for ST (11.34 vs. 11.05; the higher the better), suggesting the MS2 spectra by FLASHIda have better quality in terms of the database match score. Moreover,*

when we compared the ambiguity in the position of modification on proteoform ID reported by TopPIC, it was lower for FI spectra than ST (17.3 vs. 18.5 on average for proteoform IDs with modifications)."

4) The fact that the improvement was only 5-10% when using TopFD deconvolution seems to be an important point that did not have much discussion and maybe requires further exploration... If ~60% deconvoluted masses overlapped with a SNR of 1, what explains the low ID improvement with FLASHIda when using TopFD deconvolution? Were all of the additional IDs in the TopFD analysis part of the 40% deconvoluted proteoforms that did not overlap with the SNR threshold?

We thank the reviewer for this comment. The additional IDs in ST datasets by the TopFD analysis (Supplementary Fig. 13a) are indeed the ones with low precursor SNR values (<1.0). Thus, we applied a precursor SNR threshold of 1.0 to obtain the ID counts in Supplementary Fig. 13b. While many IDs in ST datasets (about 30%) were discarded by this filtration, we also found that a similar portion of IDs in FI datasets were also filtered out. These removed spectra in FI datasets obviously have high precursor SNR (>1.0) when FLASHDeconv (in FLASHIda) is used for deconvolution, since FLASHIda acquired only such precursors. But they have low SNR (<1.0) when TopFD is used instead. This is because precursor SNR is determined by both precursor and its deconvolved mass; when different deconvolution methods report different masses, a single precursor may have different precursor SNRs. In the opposite case, there may be precursors of high SNR with TopFD deconvolution but of low SNR with FLASHDeconv. But they are not acquired by FLASHIda. Thus, using FLASHdeconv mass deconvolution for FLASHIda acquisition ensures that the acquired precursors have high SNR values leading to high identification rate. We added the following paragraph on page 11-12:

"When the FI and ST datasets were deconvolved by TopFD14 (for both MS1 and MS2) instead of FLASHDeconv, only 5-10% more proteoform IDs were found in FI datasets than in ST datasets (Supplementary Fig. 15a and Supplementary Table 2-6). We also found that TopFD yields more proteoform IDs from ST datasets than FLASHDeconv. When the precursor SNRs of TopFD proteoform IDs in ST datasets were measured, about 30% of them had the SNRs lower than 1.0, which contributed to the increased proteoform IDs (as described above, FLASHDeconv only retains proteoforms of precursor SNRs higher than 1.0). Thus, we attributed the low ID boost in FI datasets to these low precursor SNR proteoforms and applied the precursor SNR threshold of 1.0 to all TopFD IDs (see Supplementary Fig. 16-20 for examples of low precursor SNR TopFD IDs). This filtration, as expected, removed 30% of TopFD IDs from ST datasets. But we also observed that 30% of the proteoform IDs from FI datasets were removed by this filtration,

again yielding less than 10% improvements on proteoform ID on FI datasets. (Supplementary Fig. 15b). These removed spectra in FI datasets obviously have high precursor SNR (>1.0) when FLASHDeconv (in FLASHIda) is used for deconvolution, since FLASHIda acquired only such precursors. But they have low SNR (<1.0) when TopFD is used instead. This discrepancy arises because precursor SNR is determined by not only the precursor but also its deconvolved mass; when different deconvolution methods report different masses, a single precursor may have different precursor SNRs. In the opposite case, there may be precursors of high SNR with TopFD deconvolution but of low SNR with FLASHDeconv. These are typically not acquired by FLASHIda. Thus, using FLASHdeconv's mass deconvolution for FLASHIda acquisition ensures that the acquired precursors have high SNR values leading to high identification rate. Indeed, when the precursor masses from FLASHIda were used instead of TopFD precursor masses (with MS2 deconvolution still done by TopFD), the proteoform ID counts for ST datasets almost did not change, but those for the FI datasets increased by 50-65%, recovering the boost by FLASHIda acquisition (see Supplementary Fig. 15c and Supplementary Table 7; note that in Fig. 2 and 3, FLASHDeconv was used for both MS1 and MS2 deconvolution). Since MS2 deconvolution is usually dependent on precursor mass and charge values, the boost could be even higher if an interface between FLASHIda and TopFD were implemented."

5) In line 203, the authors mention a boost being "recovered to 50-65%"... how do these results analyzed here differ from the previous paragraph that mentioned a doubling of IDs and also used TopFD and FlashDeconv as the deconvolution algorithm? Some clarification here is needed.

We hope the above response cleared this point. To add a bit more explanation, the results from Fig. 2a are obtained with FLASHDeconv deconvolved MS1 and MS2 spectra. The results in this paragraph (Supplementary Fig. 13c) are obtained with FLASHDeconv deconvolved MS1 and TopFD MS2 deconvolved MS2 spectra. We clarified that MS1 and MS2 spectra were deconvolved by FLASHDeconv in the original analysis.

6) I think the paragraphs describing the proteoform identifications under "Analysis of proteoform and protein coverage" section would be more relevant to the paper and interesting if the discussion really highlighted how FLASHIda enabled these results... for example, a few paragraphs describe the signal peptides, truncated proteoforms, and modified proteoforms identified in FLASHIda, but it would be interesting to know if these were also identified in the standard runs? If they were identified in standard runs, this discussion is not very relevant to the paper; if these proteoforms were not identified in the standard run, this fact should be much more

highlighted as an interesting result and strength of the platform to reveal additional proteoform biology.

We thank the reviewer for suggesting this, which greatly improves our manuscript. To address this comment, we emphasized the part in which FLASHIda gives more findings than standard DDA (e.g., in Fig. 3c, FLASHIda has many exclusive jointly detected proteins as compared with standard DDA). Moreover, we added more comparative analyses like GO term analysis and truncated protein analysis. The modified and added paragraphs are found in the 'Analysis of proteoform and protein coverage' section on page 13-15 (blue font paragraphs and sentences). We would rather avoid including all changes in this response since they are rather scattered over the manuscript.

7) In general, I think the paper needs better discussion and analysis of additional proteoform IDs (in the doubling ID count for standard vs. FLASHIda) - for example, were these proteoforms selected for fragmentation in the standard runs at all? Were they observed in the standard runs but not selected?

We performed the suggested analysis and a few more relevant analyses, and added the following paragraph on page 9.

“While the ST datasets have almost 20% more precursors acquired than in the FI dataset (40,643 in ST vs. 32,997 in FI), the number of deconvolved precursors in ST datasets are almost 30% fewer than in FI (25,579 in ST vs. 32,702 in FI). The precursors of high precursor SNRs were more than twice as large in the FI than in the ST dataset (14,255 in ST vs. 32,208 in FI). Out of these precursors, more than twice as many distinct nominal masses were acquired for FI than for ST (1,924 in ST vs. 4,592 in FI), demonstrating that FLASHIda results in precursors of far diverse masses and of higher quality than standard DDA. When we compared the proteoform IDs from each acquisition, 1,774 nominal proteoform ID masses were exclusively identified in FI datasets, and only 105 (5.9%) masses out of these exclusive masses were acquired and passed precursor SNR filtration (but not identified) in the ST datasets. On the other hand, out of 217 exclusive nominal proteoform ID masses in the ST datasets, 100 (46.1%) were acquired but not identified in the FI datasets. Per each proteoform ID, the average numbers of acquired precursors were 4.8 for FI and 7.0 for ST, demonstrating a less redundant precursor selection of FLASHIda. In contrast, the average numbers of identified MS2 spectra per proteoform ID were 3.5 for FI and 3.2 for ST; thus, FLASHIda acquired fewer spectra, but identified more MS2 spectra per proteoform ID than standard DDA.”

Since there is no experimental difference between FLASHIda acquired and standard DDA acquired MS1 datasets, most proteoform ions in FT are expected to appear in ST datasets.

Were they fragmented in the standard runs but only once due to the simple mass exclusion list used?

For the exclusion, we used m/z exclusion rather than mass exclusion (called charge state exclusion) based on the below observation by Dupre et al. 2021, where detailed analysis for different acquisition parameter sets were investigated with *E. coli* lysate and other bacterial samples:

“We then enabled the mass spectrometer to select and fragment only a single charge state per protein in order to reduce the MS/MS information redundancy and increase the number of identifications. Note that in that case, only determined charge states are selected for fragmentation. A significant decrease of fPrSM and informative MS/MS spectrum was observed, with lower identification scores, although the number of proteins and proteoforms identified was found almost identical to the best previous experiment”

Since m/z exclusion was employed, ST datasets have repeated mass acquisition as stated above. However, even if standard DDA uses its own mass exclusion (based on their deconvolution), there can be repeated masses due to deconvolution errors or deconvolution inconsistency between theirs and FLASHDeconv.

Additionally, finding a few examples that validate the claim that FLASHIda enabled identification of proteoforms that would not be identified by standard data acquisition would be helpful, such as for a few of the biologically interesting proteoforms discussed in comment #6.

As we wrote above, FLASHIda runs yielded many exclusive proteins and proteoforms. Also as reported in Fig. 3e-f, FLASHIda enabled the detection of more modifications than standard DDA. More analyses are found on page 13-16 (blue font paragraphs and sentences):

Page 13:

“In the ST90 dataset, only 35 proteoform IDs had masses larger than 30 kDa. The dynamic range of the proteoform intensities (measured by the feature area of each proteoform ID; different from precursor intensity dynamic range in Fig. 2b) was about five to six orders of magnitude for both FI90s and ST90s datasets, but FI90s reported far more proteoform IDs with lower proteoform intensities, showing the proteoform ID count boost

by FLASHIda is mostly achieved for low abundance proteoform species (see Supplementary Fig. 22)."

Page 14:

"In ST90s datasets, however, far fewer biological process GO terms were reported (96 in FI90s vs. 59 in ST90s), and the energy-related terms were not reported (see Supplementary Table. 9). The molecular function GO analysis results from FI90s and ST90s were almost identical. We can thus confidently state that FLASHIda is able to provide rich information even for families of proteoforms including low-abundance species."

Page 15:

"Fig 3e and 3f also show that the number of modified proteoform IDs in the FI datasets are almost twice those of the ST datasets (similar to unmodified proteoform IDs). Moreover, when the same frequency threshold (>10% the most frequent mass shift, which is oxidation) for the modification annotation was applied, FI datasets yielded exclusive annotations such as methylation and replacement of a proton by potassium."

Page 15-16:

"In addition to them, many putative proteins with signal peptides have been identified in FI and ST datasets, including many FI exclusive ones. For instance, when we collected only unmodified proteoform IDs (to remove possible false positives) with the truncation of the N-terminal 16-30 amino acids, 50 and 37 proteoform IDs were found from FI90 and ST90 datasets, respectively. Out of them, 12 were FI90 dataset exclusive, containing ribose import binding protein RbsB (UniProtKB: P02925) with known signal peptide cleavage. In total, 600 unmodified and truncated proteoforms have been identified from the FI90 dataset while 375 have been identified from the ST90 dataset. Thus we can conclude that FLASHIda enables more sensitive detection of proteins and proteoforms not seen with standard acquisition in complex sample studies."

Reviewer #2

Overall, I find this work to be very well done and it is a welcomed addition to the field of top down proteomics. The incorporation of online FLASHDecon and the Qscore approach to precursor selection is novel and shows a marked improvement in both protein and proteoform identification. The C# source code is well written and well documented. I did feel, however, that the discussion around Figure 3B and 3D was superfluous and not necessarily supportive of FLASHida. I am also concerned that the noteworthiness of this work is limited in light of past works of Durbin, Fornelli, and Lu.

We truly appreciate the reviewer's positive comment on our manuscript. Fig. 3b shows how proteoform ID masses elute along retention time for FLASHida acquired datasets. This figure could be informative to the readers by providing the overall mass and dynamic ranges and the uneven distribution of identified masses along retention time and mass range. From this uneven distribution, we also could see the need for different selection strategies for different retention time and mass values. In this regard, we added the following sentences on page 16 in the Discussion section:

“For instance, from the uneven distribution of masses in Fig. 3b, one can find the need for different selection strategies for different proteoform ion mass or RT ranges.”

In addition, the eluted ions in the deconvolved mass axis could be more informative than in m/z to see, for example, whether the identified masses were eluted within the expected range (e.g., small masses tend to elute earlier and larger later in our LC setup).

Fig. 3D shows the GO term analysis for FI90s datasets, and as the reviewer pointed out, it has limited information as is. Thus we added GO term analysis for ST90s datasets in the Supplementary Table. 9 and put additional analysis on page 13-14 (blue font paragraphs and sentences). We skip putting the sentences in this response since they are rather scattered over the manuscript.

Regarding to the novelty of this work:

While both AutoPilot and MetaDrive use the concept of real-time processing of spectra, both works clearly pointed out the limitation caused by the long processing time. The rapid deconvolution algorithms unique to FLASDeconv/FLASHida address this major bottleneck in both approaches.

The authors of AutoPilot specified that the low proteoform identification count was due to narrow search space for AutoPilot real-time identification and clearly stated that “This search strategy was chosen to sufficiently limit the search space to enable searches to

be performed on the same desktop computer that controls acquisition and finish within the cycle time of the instrument.”

In the case of MetaDrive, the authors were able to increase the proteoform identification rate by 10% by their acquisition method in yeast samples. But the increase in the proteoform ID count was limited due to the long deconvolution processing time, as the authors clearly stated that “the computation time for the real-time charge deconvolution algorithm is a source of delay in the decision-making process.” Thus, our work has high value in that FLASHIda overcomes this unsolved hurdle by leveraging fast deconvolution of FLASHDeconv. In this regard, we added the following paragraph in Introduction section on page 4:

“Both methods used real-time mass deconvolution (i.e., real-time determination of intact proteoform masses) to disentangle complex signal structure of proteoforms. Autopilot also employed real-time database search for better characterization of proteoforms as well as for less redundant precursor mass selection, while MetaDrive focused on improving the quality of the fragment spectrum by dissociating multiple precursor ions of the same proteoform. Both approaches clearly demonstrated the potential of IDA for TDP studies and, at the same time, were limited because of the need for faster processing times that could fit within the cycle time of the MS instrument.”

1. (Line 63): *Is mass-based exclusion not common on modern instruments? I was under the impression that this was now standard during top down acquisition.*

The standard acquisition parameters were chosen according to the recommendation in Dupre et al, J Proteome Res. 2021, where detailed analysis for different acquisition parameter sets were investigated with *E. coli* lysate and other bacterial samples. For the exclusion, we used m/z exclusion rather than mass exclusion (called charge state exclusion) based on the below observation by Dupre et al. 2021:

“We then enabled the mass spectrometer to select and fragment only a single charge state per protein in order to reduce the MS/MS information redundancy and increase the number of identifications. Note that in that case, only determined charge states are selected for fragmentation. A significant decrease of fPrSM and informative MS/MS spectrum was observed, with lower identification scores, although the number of proteins and proteoforms identified was found almost identical to the best previous experiment”

2. (Line 67): *I feel that the range of 5-15 Th for an isolation is a bit dated; recently released instrumentation commonly use a quadrupole to isolate in the 1.2 - 3 Th range. Lubeckyj et al. (18) mentioned a window of 4Th and Toby et al. (5) mentions*

15Th only as 'ultra-wide isolation'. Isolation width is very important in proteoform discovery, but I don't think this approach improves on the current state of the art in this way. Line 462 mentions that you used a width of 3 Th.

We rephrased the sentence to “relatively high isolation window size ranging from 1.2 to even 15 Th (ultra-wide isolation).”

The 3 Th isolation window size is only for ST datasets. We clearly rephrased the sentence to “3 Th isolation window for ST datasets and dynamic isolation window size for FI datasets.” We appreciate the reviewer’s detailed comment.

We also added additional sentences at the end of FLASHIda overview section on page 5 as well as in Figure 1b legend:

“Lastly, the charge state and isolation window size for each selected mass are dynamically determined to minimize interference from noise or coelution. The determined isolation window m/z ranges are provided to the instrument through the Thermo iAPI interface.”

Lastly, how the isolation window size is determined by FLASHIda is added on page 7-8 as follows:

“For standard, a fixed isolation window size of 3.0 Th was used for all runs. For FLASHIda runs, the isolation window size is dynamically controlled so the isolation window covers the complete precursor envelope range with 0.6 Th margin from both sides. On average, the isolation window size from FI datasets was 3.1 Th.”

3. (Line 74): *Two crucial citations were missing that call into question some assertions made in this section. Given the age of Durbin et al., a follow up "Autopilot" search turned up 2 follow up papers: "Quantitation and Identification of Thousands of Human Proteoforms below 30 kDa", Durbin et al. 2016 and "Advancing Top-down Analysis of the Human Proteome Using a Benchtop Quadrupole-Orbitrap Mass Spectrometer", Fornelli et al. 2017.*

We added the citations as suggested. Also we added the following sentence to remark that our quality-based exclusion scheme is similar to the one used in Autopilot on page 6 last paragraph:

“Similar to the exclusion by the confidence of the identification used in Autopilot, FLASHIda only excludes masses that are highly likely to be already identified”.

4. *There were a couple word choices that I would change. What does 'divertize' mean (lines 71 and 675)? Please revise the word 'instant' to be more quantitative (line 93).*

We rephrased the first sentence as “To reduce the redundancy in precursor ion selection.” We replaced the “instant mass deconvolution” expression with “mass deconvolution within 10-50 ms.”

5. *I really liked how you structured your logistic regression (starting on line 98) and how thoughtfully you laid out the six features (Methods).*

We truly appreciate the reviewer’s kind comment.

6. *(Line 104): Could you comment more on the preliminary studies you conducted to determine features?*

We added the following paragraph in the Methods section page 27:

“To determine informative features, we first included only three features, charge_cos, charge_dist, and mass_cos as they are directly available from FLASHDeconv. In 10-fold cross validation, the trained model resulted in ROC (Receiver operating characteristic) area of 0.775. However, we found that only with these features, coeluted precursors or harmonic masses are not penalized enough. Thus we added two more SNR related features: precursor_SNR and mass_SNR. The resulting ROC area increased to 0.79. Lastly, we added mass_error achieving the ROC area of 0.80. The addition of peak or mass intensity features almost did not improve the classification performance (ROC area of 0.80). But precursor mass and charge related features improved the classification (ROC area of 0.87). This boost, however, introduced a strong bias towards smaller proteoform masses (<20 kDa) resulting in drastically reduced number of heavy proteoform IDs. Moreover, the optimal weight for the precursor mass or charge feature may be strongly dependent on instrumental settings, e.g., resolution, making the model less robust than without it. Thus we finalized the six features listed as above.”

Also, while revising the manuscript we found that this feature list is quite similar with the list used in DScores introduced in Marty et al., 2020. Thus we added this reference on page 6.

7. *It was mentioned that the TQScore threshold was 0.9 (lines 130, 147) and the QScore threshold was 0.25 (line 148). How were these arrived at?*

We determined the thresholds based on our manual observation and empirical analysis. When TQScore is set to a value which is too high (e.g., 0.95), the selection of precursors becomes rather redundant resulting in reduced proteoform IDs. In the case of QScore, we observed that most precursors of QScore lower than 0.25 have poor signal shape when seen manually. We will need statistical methods to determine better thresholds. We added this point to the Discussion section on page 17-18 as follows:

“In addition, the QScore and TQScore thresholds (0.25 and 0.9, respectively) used in this study were determined rather empirically and manually; statistical methods to determine optimal thresholds could yield most proteoform IDs for different setups.”

8. (Line 131): *I'm not sure what '(of proteoform)' means? Perhaps a typo?*

We changed this into (of proteoform ion).

9. (Line 147): *400 Da seems very low for a mass cutoff. Given your scan range was 500-2000 Th (line 461) and your lower charge limit was 4 (line 464), wouldn't 2000 Da be a more appropriate limit?*

We agree with the reviewer's comment. Those parameter values have been determined rather asynchronously; charge and mass ranges are from FLASHIda software side and scan range is from machine side. In the future, more interactive information sharing is needed between FLASHIda and the instrument.

10. (Line 167): *It is notoriously difficult to download a specific version of a UniProt database after the fact. Please consider including an archived fasta file in your supplemental data.*

We added the used fasta file (Supplementary File 1).

11. *I appreciate that you were careful in how you reported protein and proteoform counts (around line 169). Choosing to compare the average across the technical replicates, and not some naïve "total" count, preserves control of the FDR.*

Again, we deeply appreciate the reviewer's kind remark.

12. (Figure 2A): *Why does the ST90s bar on # proteoform IDs not match Supplemental Figure 13c?*

The results from Fig. 2a are obtained with FLASHDeconv deconvolved MS1 and MS2 spectra. The results in Supplementary Fig. 13c are obtained with FLASHDeconv deconvolved MS1 and TopFD MS2 deconvolved MS2 spectra. We clarified that MS1 and MS2 spectra were deconvolved by FLASHDeconv in the original analysis for Fig. 2a.

13. *(Figure 2B, right): I noticed that the ST experiment (top) targets a small population of higher intensity precursors (i.e., the histogram on the ST chart goes farther to the right than the histogram for the FI chart). Can you speculate as to why this is the case?*

We appreciate the reviewer's insightful observation. The precursor prioritization in the standard acquisition is mostly intensity-based while that in FLASHIda is quality (QScore)-based. Thus it is expected that the precursors in ST have higher intensities than in FI overall. However, we also can observe that FI precursor intensity dynamic range is higher than ST, which means FLASHIda enables the selection of low intensity but high quality precursor selection. Also Figure 2b left shows that FI precursors have higher quality (QScore) than ST on average. We added the following sentences on page 11:

"Comparing precursor intensity distributions (Fig. 2b, right panel) reveals that FI precursors have lower intensities than ST precursors on average. This is because precursor prioritization in the standard DDA acquisition is mostly intensity-based while that in FLASHIda is quality (QScore)-based. However, FLASHIda achieves almost one order of magnitude higher dynamic range than standard DDA, suggesting that FLASHIda enables the selection of low-intensity, but high-quality precursors."

Also, newly added Supplementary Fig. 22, we pointed out that the proteoform ID count boost by FLASHIda is mostly achieved for low abundance proteoform species (see comment 15 below).

14. *I don't think the claim of the subsection title "IDA improves reproducibility of proteoform level identification" on line 206 is fully verified. While I fully accept that you found more proteoforms, on line 212, the reproducibility comparison is described only as "comparable". Please make the improvement in reproducibility clearer or tone down to the claim of the subsection.*

We agree with the reviewer's opinion and changed the subsection title to "IDA increases the number of overlapping proteoform level identifications" and also added a sentence describing the increased overlapping proteoform IDs by FLASHIda acquisition on page 12.

15. In the section "Analysis of proteoform and protein coverage" starting on line 221, I feel that a comparison of FI and ST should be included. For example, did the ST90 proteoforms over 30kDa outperform the McCool study? The dynamic range of proteoform intensities for FI90 is 5-6 orders of magnitude; how does this compare to ST90?

In the ST90 dataset, only 35 proteoform IDs had larger masses than 30 kDa (as compared to 105 in FI90). In the case of proteoform intensity dynamic range, FI90s and ST90s showed comparable ranges (apart from precursor intensity dynamic range in Fig. 2b). But FI90s found far more IDs with low intensities. We modified the paragraph on page 13 as follows and added the figure showing the histogram of proteoform intensities for FI90s and ST90s in Supplementary material (also added below).

"In the ST90 dataset, only 35 proteoform IDs had masses larger than 30 kDa. The dynamic range of the proteoform intensities (measured by the feature area of each proteoform ID; different from precursor intensity dynamic range in Fig. 2b) was about five to six orders of magnitude for both FI90s and ST90s datasets, but FI90s reported far more proteoform IDs with lower proteoform intensities, showing the proteoform ID count boost by FLASHIda is mostly achieved for low abundance proteoform species (see Supplementary Fig. 22)."

Supplementary Figure 22. The histogram of $\log_{10}(\text{proteoform intensity})$ for FT90s and ST90s datasets. The proteoform intensity is measured by the mass feature area of each proteoform ID reported by FLASHDeconv. It is clearly shown that the proteoform ID boost by FLASHIda is mainly obtained from low intensity proteoforms.

In addition, we added more comparative analyses between FLASHIda and standard DDA on pages 13-16.

Page 13:

“In the ST90 dataset, only 35 proteoform IDs had masses larger than 30 kDa. The dynamic range of the proteoform intensities (measured by the feature area of each proteoform ID; different from precursor intensity dynamic range in Fig. 2b) was about five to six orders of magnitude for both FI90s and ST90s datasets, but FI90s reported far more proteoform IDs with lower proteoform intensities, showing the proteoform ID count boost by FLASHIda is mostly achieved for low abundance proteoform species (see Supplementary Fig. 22).”

Page 14:

“In ST90s datasets, however, far fewer biological process GO terms were reported (96 in FI90s vs. 59 in ST90s), and the energy-related terms were not reported (see Supplementary Table. 9). The molecular function GO analysis results from FI90s and ST90s were almost identical. We can thus confidently state that FLASHIda is able to provide rich information even for families of proteoforms including low-abundance species.”

Page 15:

“Fig 3e and 3f also show that the number of modified proteoform IDs in the FI datasets are almost twice those of the ST datasets (similar to unmodified proteoform IDs). Moreover, when the same frequency threshold (>10% the most frequent mass shift, which is oxidation) for the modification annotation was applied, FI datasets yielded exclusive annotations such as methylation and replacement of a proton by potassium.”

Page 15-16:

“In addition to them, many putative proteins with signal peptides have been identified in FI and ST datasets, including many FI exclusive ones. For instance, when we collected only unmodified proteoform IDs (to remove possible false positives) with the truncation of the N-terminal 16-30 amino acids, 50 and 37 proteoform IDs were found from FI90 and ST90 datasets, respectively. Out of them, 12 were FI90 dataset exclusive, containing ribose import binding protein RbsB (UniProtKB: P02925) with known signal peptide cleavage. In total, 600 unmodified and truncated proteoforms have been identified from

the FI90 dataset while 375 have been identified from the ST90 dataset. Thus we can conclude that FLASHIda enables more sensitive detection of proteins and proteoforms not seen with standard acquisition in complex sample studies.”

16. In the section discussing the GO analysis (lines 239-61), I feel that comparisons between FI and ST are needed. As it stands, this section only serves to validate that your workflow matches a previous BUP study and does little to bolster FLASHIda vs. 'standard' top down.

We agree with the reviewer's point. We performed GO term analysis for ST90s datasets and included them in Supplementary Table 9. Indeed we found that GO terms reported in FI datasets are more relevant and informative than in ST datasets. We added the following paragraph on page 14:

“In ST90s datasets, however, far fewer biological process GO terms were reported (96 in FI90s vs. 59 in ST90s), and the energy-related terms were not reported (see Supplementary Table. 9). The molecular function GO analysis results from FI90s and ST90s were almost identical. We can thus confidently state that FLASHIda is able to provide rich information even for families of proteoforms including low-abundance species.”

*17. (Line 480): Is it correct to say that FLASHIda uses an isolation window of 1.2 Th ($2 * 0.6$ Th) and the standard workflow uses an isolation of 3 Th (line 462)? If so, why the difference? My concern is that this difference might allow FI to pluck out specific proteoforms where ST would have found co-isolating chimeras (and been filtered).*

The isolation window covers the complete precursor envelope range with 0.6 Th margin from both sides. This means the isolation window size is dynamically determined by the 1.2Th plus precursor envelope range of each target precursor mass. On average, the isolation window size was 3.1Th for FI datasets. We clearly state this point in the revised manuscript to avoid misunderstandings (page 8 in main text and page 25 in Methods section).

18. (Line 483): Is this saying that you have 2 MS1 scans between each fragmentation scan? My understanding from the rest of the paper is that you have only a single MS1 scan (e.g., Figure 1A).

We have one survey MS1 scan (high-resolution Orbitrap scan used for the analysis) and one AGC scan (low resolution ion trap scan used to assess the ion flux) after every **set** of fragmentation scans. Although the AGC scan is also an MS1 scan its sole purpose is to assist the Automatic Gain Control function of the mass spectrometer. Usually, AGC scans are performed by all trapping instruments silently (they are not written into the result file), however, we needed to perform them explicitly. The latter is the limitation of the employed IAPI version and might be changed in the future versions.

19. (Line 555): It seems that the extreme charges, w and W , are left out of the penalty calculation. Is this correct? If so, why?

The charge distribution score observes the intensity relation between consecutive charges. Even if the formulas seem to exclude the extreme charges, they actually consider all possible pairs of consecutive charges. For example, the left penalty is given by $l_z = \max(0, i_{z-1} - i_z)$, for $z=w+1, \dots, x$. When we plug in z values into the l_z definition, we can see that it starts with the intensity difference between i_w and i_{w+1} , including the smallest charge w .

20. In the C# source code, there is mention of a "magic scan identifier" of 41. What is this and why is it needed?

The sole purpose of a "magic scan" is to detect that FLASHIda has control over the instrument and, thus, can start execution of the method. When using IAPI, it is possible to set a special property of a scan requested from the instrument - Access ID. "Magic scan" is just a scan with Access ID set to 41 (the value is arbitrary, as soon as it does not equal 0 or 1, since these values are used by the instrument internally). The software requests this magic scan from the instrument and as soon as it receives it back, it "knows" that the instrument is ready to receive further commands.

21. Are you planning to open source the code for the Qscore? I didn't see it in the C#.

The QScore calculation is part of the FLASHDeconv source code which is found in www.OpenMS.de/FLASHDeconv, which is used to determine the features weights.

22. As noted in the summary, I wanted to reiterate that the C# source code and documentation is well done and goes above and beyond what is typical. I enjoyed looking through it and plan to investigate the DataFlow library for some of my projects!

We appreciate the reviewer's kind comment on documentation. As FLASHIda is an open source tool, thus please feel free to use its source code for your projects. And in the case you need any help, we will gladly provide any possible support.

23. *(Supplemental Figure 12): The caption says 2kDa, I believe it should be 20kDa.*

Thank you for spotting this mistake. We corrected it in the revised manuscript.

Reviewer #3

Development of the smarter data acquisition methods in mass spectrometry-based proteomics is one of the most important issues, particularly for top-down proteomics. This is mainly due to the diverse MS1 signals of different charge states from the same proteoform. In this manuscript Kyowon et al. present an improved IDA algorithm, FLASHIda, to resolve the problem of low coverage of proteoforms. FLASHIda highly relies on FLASHDeconv, which is a fast algorithm on spectral deconvolution developed by the authors' lab. Much of the studies in this new manuscript went to algorithmic development to assess the quality of the masses (protein species in MS1) and then selected top-N high-quality masses to trigger MS2 fragmentation according to their T-Qscores. Of all the experiments results presented, the most impressive one is the count of the identified proteoforms doubled by FLASHIda when compared it to the standard DDA method.

In one word, the authors have presented an effective IDA algorithm and significantly improved the identification rate of top-down MS/MS. In order to improve the submitted manuscript further, the following suggestions to include more informative and convincing data results are listed.

Major:

1. In figure 2a, the number of precursors identified in FI90s is about 6000. The number of proteoforms identified in the same data set is about 1500. Therefore, each proteoform is identified four times on the average. Is this right? If it is right, why for the same proteoform, it is identified repeatedly?

The reviewer's point is correct. The precursor masses corresponding to the same proteoform are often selected by FLASHIda mostly to ensure that at least one of the selected precursors (i.e., MS2 spectra) is successfully identified. Also, in some retention time range, FLASHIda finds only a few available precursor masses (as shown in Fig. 3b). In this case, FLASHIda selects the same mass repeatedly. This point was already described in the manuscript (page 6-7) as follows. We supplemented the paragraph with one more sentence (bold font) as follows:

"We found that the number of high quality proteoform precursors at a given RT is often very low as compared with that of peptide precursors in BUP. Therefore, the use of the naive mass exclusion alone resulted in the selection of low quality masses, and in turn an overall drop of identification rate (see Supplementary Fig. 1). Similar to "the exclusion by the confidence of the identification" used in Autopilot, FLASHIda only excludes masses that are highly likely to be already identified. To this end it keeps a list of the triggered

*masses over a short RT duration along with their QScores. If a mass has been acquired multiple times, the probability that at least one of the acquired MS2 spectra from the mass is identified is calculated with its QScores (see Methods for details). Note that this probability, called TQScore (Total QScore) of the mass, is the probability that the proteoform of the mass is identified. After each acquisition, TQScores are immediately updated. If the TQScore of a mass exceeds 0.9, the mass is registered in the exclusion list for a short RT duration. **If all deconvolved precursor masses are already registered in the exclusion list, FLASHlda selects the one with the lowest TQScore.** Using this quality-based mass exclusion list, FLASHlda assures each triggered mass (of proteoform ion) is identified while still maximizing the diversity of proteoforms.”*

So, the comparison with DDA on the number of repeated identified proteoforms is suggested to be analyzed. That means for the E. coli data sets, on average, how many times MS/MS identified for each proteoform under FI90s and ST90s?

Per each proteoform ID, the average numbers of acquired precursors were 4.8 for FI and 7.0 for ST, again showing far less redundant precursor selection of FLASHlda. In contrast, the average numbers of identified MS2 spectra per proteoform ID were 3.5 for FI and 3.2 for ST; per proteoform ID, FLASHlda exhibited less acquired but more identified MS2 spectra than standard DDA, clearly demonstrating FLASHlda selected precursors are far more likely to be identified. We added this analysis on page 9 (second paragraph), followed by deeper analysis on why FLASHlda results in increased proteoform ID count (on page 9-10). Also please check page 13 showing that the proteoform ID count boost by FLASHlda is mostly achieved for low abundance proteoform species.

2. As we know, for most highly abundant proteoform, if it is not co-eluted and also with much less interference from other proteoforms, its experimental isotopic envelope will be well fitted to the computational one. However, in highly complex samples, their top-down data will not be so ideal. The co-eluted, the interference, and the low abundant skewed distribution will be everywhere. How to deal with those proportion more effectively?

We thank the reviewer for bringing this issue up. The original manuscript already pointed out this issue. The reason why we came up with the notion of “precursor SNR” is to remove or reduce the selection of precursors possibly containing multiple proteoform ions (resulting in so-called chimera MS2 spectra). We rephrase the manuscript so the definition of precursor SNR is more clear as follows on page 8:

“Lastly, FLASHlda also discarded precursors of SNR within the precursor envelope m/z range (called precursor SNR; see Methods) lower than 1.0. Briefly, precursor SNR is

defined by the power of targeted peaks divided by that of non-targeted peaks within the precursor envelope m/z range. When coeluted ions are present within the m/z range (resulting in so-called chimera spectra) or the precursor mass represents mass artifacts (see examples in Supplementary Fig. 2-6), the power of the non-targeted peaks increases and in turn the precursor SNR decreases. Since the chimera spectra and the spectra of incorrect precursor masses often cause target-decoy based false discovery rate (FDR) control inaccuracies and even erroneously inflate the number of proteoform IDs (see Methods, Supplementary Fig. 7-8 and Supplementary Table. 1 for detail), the precursors of low precursor SNR were avoided in data acquisition.”

We also added additional sentences in the Methods FLASHIda overview section on page 33 to emphasize FLASHIda effectively deals with possible interference:

“Lastly, the charge state and isolation window size for each selected mass are dynamically determined to minimize interference from noise or coelution. The determined isolation window m/z ranges are provided to the instrument through Thermo iAPI interface.”

For example, if there are two different proteoforms in the same isolation window, the peaks not included in one isotopic clusters are not necessarily noisy peak, but possibly from another proteoform. This will bias the SNR estimation if we ignore any one proteoform. Furthermore, how to assess their quality by Q-score? This phenomenon is not atypical in typical top-down data sets.

We are well aware of this problem of chimera spectra in TDP studies and introduced the notion of precursor SNR, as stated above. The notion of precursor SNR considers not only the signal power from the target ion but also the noise power from other ions, within the isolation window. If coeluting proteoform ions exist within the isolation window, the corresponding peaks are interpreted as a noise component and in turn decrease precursor SNR. We believe the above paragraph on page 8 clearly states this point.

3. For those high-quality acquired masses with MS/MS, there are still one quarter (25%) unidentified? what are the main reasons? Perhaps the one unknown PTM in TopPIC setting is the reason to lose those proteoforms with multiple modifications. Another is the FDR cut-off. So a simple analysis for all those 25% unidentified will show some clues to improve the ID rate further. Searching those reasons is not a simple task because we do not the answer for some. However, we can also know some, such as listed above.

FLASHIda acquires precursors of QScore higher than 0.25. Since the QScore of a precursor is an estimated probability that the acquired MS2 from the precursor is identified, not all MS2 spectra have extremely high chances to be identified. Moreover, QScore estimation may not always be accurate (now as described in page 17 second paragraph). Thus, it is expected that a certain portion of MS2 spectra are not identified. We also examined whether identified MS2 spectra from FI datasets and those from ST datasets have quality differences, but the difference was not very high. When we simply take two sets of all identified MS2 spectra (one from FI and the other from ST datasets), their quality differences (in terms of e-value, the number of annotated fragments, and the PTM localization ambiguity) were negligible. Next, for each FI and ST dataset, we collected 1,171 MS2 spectra from the jointly identified proteoforms (found in *proteoform.tsv output files by TopPIC). For these MS2 spectra, the average value of $-\log_{10}(\text{evalue})$ was higher for FI datasets than ST datasets (11.34 vs. 11.05). Moreover, the PTM localization ambiguity was lower for FI than ST (17.3 vs. 18.5). This shows that for the same proteoform precursors, FLASHIda yields slightly better quality MS2 spectra than standard DDA.

4. Most proteoforms are modified by PTMs as shown in Figures 3e and 3f. In figure 3f, please replace the integer mass shift by the accurate mass with at least four or six digits because some modification masses are the same when only use integer mass. What is the frequency of all those listed PTMs found according to the ID results? Please give another column to show their appearance frequency or counts. For figure 3e, please annotate the accurate mass on those peaks.

We used a bin size of 0.05 and annotated the mass values for the major mass shifts in Figure 3e as the reviewer suggested. Also we specified the numbers of the mass shifts in Figure 3f for both ST and FI datasets. The figure legend for Figure 3 has been modified accordingly.

5. In top-down data analysis by averagine model, the 'One-dalton' Shift is a troublesome question. This means there will be one-dalton or two-dalton mass shift between the prediction by averagine model and the accurate monoisotopic mass. This is validated by the results after identification of their MS/MS. So in the E coli data sets of FI90s, please statistically analyze the one-dalton shift among all those precursors identified, such as what is those percentages, etc.

We thank the reviewer for this excellent comment. Frequent one Da mass error could be one of the reasons for increased proteoform ID count in FI datasets. To see if this is the case, we drew the mass error histogram in Supplementary Fig. 10 and the one Da mass error ratio along proteoform mass in Supplementary Fig. 11 (attached below). Overall,

FLASHIda generated IDs showed a comparable ratio of one Da mass error (7.2% and 9.9% in ST and FI datasets, respectively). Moreover, for larger proteoform IDs, FLASHIda showed less error ratio. We added the following paragraph on page 10:

“To see if the increased unique precursor mass count is due to the frequent one Da mass error arising from inaccurate deisotoping in precursor mass deconvolution²⁹, we examined the mass difference between deconvolved precursor mass and proteoform mass determined by identification (with Precursor mass and Adjusted precursor mass columns in TopPIC output tsv files). Standard DDA and FLASHIda generated comparable ratios of one Da mass error (7.2% and 9.9% in ST90s and FI90s datasets, respectively). Moreover, for heavy proteoform IDs (>25 kDa), FLASHIda showed less error ratio than standard DDA (see Supplementary Fig. 10-11).”

Supplementary Figure 10. The histogram of mass difference between precursor and proteoform ID for FI90s and ST90s datasets. To see if the increased unique precursor mass count is due to common one Da mass error arising from inaccurate deisotoping in precursor mass deconvolution, we examined the mass difference between deconvolved precursor mass and proteoform mass determined by identification in FT90s (upper) and ST90s (lower panel) datasets. The x-axis shows proteoform ID mass values (Adjusted precursor mass column) subtracted by deconvolved precursor mass values (Precursor mass column in TopPIC output tsv files).

Standard DDA and FLASHIda generated comparable ratios of one Da mass error (7.2% and 9.9% in ST90s and F190s datasets, respectively), where one Da mass error is defined to occur when the absolute value of the mass difference in x-axis is within 10 ppm mass tolerance.

Supplementary Figure 11. One Da mass error rates along proteome masses for F190s and ST90s datasets. The one Da mass error (defined in Supplementary Fig. 10) ratio is drawn for different ranges of proteome ID masses. The ratio (in %) is defined by (#proteome IDs with one Da mass errors divided by #all proteome IDs) per mass range. FLASHIda showed less error ratio than standard DDA for heavy proteome IDs (>25 kDa).

Minor

1. Line 40, the reference of 1 and 2 for proteoform is not the original paper. Please instead use the following original citation:

Proteoform: a single term describing protein complexity

Lloyd M Smith, Neil L Kelleher & The Consortium for Top Down Proteomics *Nature Methods* volume 10, pages 186–187 (2013)

Thank you for pointing this out - we included the correct reference in the revised manuscript.

2.L59, 'inadequate'?, perhaps this sentence can be restated as, these criteria are suboptimal for high-quality proteoform ion selection.

We have corrected this as suggested (page 3).

3.L276, the alternative start codon can also be a reason?

Indeed, this could be another possible reason. We added this reason in the manuscript (page 15 second paragraph).

4. Figure 2a, the right # precursors, the black, gray legend is not obvious to the bars below, which one is which one?

We thank the reviewer for this detailed comment. For each dataset, the grouped bars from lightest to darkest show the total, deconvolved, precursor SNR applied, and identified precursor counts. The (+) markers show the numbers from the replicates, and the bars show their average values (We added precursor SNR applied precursor counts in this revision). We gave more transparency distinction to the bars and added the above explanation in the Figure legend.

5. L563, for the negative ion mode, most precursors are deprotonated, so it is not electron. It is one proton lost for negative one charge.

Thank you for pointing this out. We changed the text as follows:

“For a charge z peak of isotope index n , its monoisotopic mass p is calculated by $p=z(t - c)-n\delta$ for positive MS or $p=z(t + c)-n\delta$ for negative MS, where t is the m/z of the peak, c denotes the mass of proton, and δ the mass difference between ^{13}C and ^{12}C .”

REVIEWERS' COMMENTS

Reviewer #1 (Remarks to the Author):

I found the reviewer responses to be well done and all of my comments were addressed.

Reviewer #2 (Remarks to the Author):

Thank you so much for your thoughtful and thorough response to my comments. Generally, I am satisfied with your changes and appreciate the additional citations and supplemental materials. Your section around the GO annotations is much improved and supportive of your conclusions. My only concern that remains revolves around my first question: why was m/z exclusion used instead of mass exclusion? Regardless of Dupre 2021, in order to compare your mass exclusion approach to the current state of the art, I would expect one to try the built-in mass exclusion list on the instrument (made all the better if Advanced Peak Determination, APD, is enabled). It is very possible that the conclusion of Dupre would come to pass, but I think the extra experiment is worthwhile.

Reviewer #3 (Remarks to the Author):

The authors have carefully addressed all issues or suggestions and also provided a detailed file to explain in details how to deal with those issues.

I agree with the authors' technical improvement on their manuscript.

Only one point or advice on Page 22 of the rebuttal file which is how to improve the identification of the chimera spectra. The current version of FLASHida uses a precursor (targeted) SNR to control the acquisition. If there are two proteoforms within one isolation window, then they will be co-fragmented. Although they are difficult to be identified when compared to species with a higher precursor SNR, they are still possible to be identified if we can identify those two precursors, such as identifying them by their corresponding isotopic envelopes. Furthermore, the proportion of chimera spectra is very high in real-life complex samples. So my concern or suggestion is we should consider these proteoforms when it is required to determine their acquisition possibility. I hope maybe in the

next version, the authors can further improve the performance of FLASHIda by increasing the number of IDs from this type of proteoforms.

REVIEWERS' COMMENTS

Once again, we thank the reviewers' constructive and sharp comments that significantly improved the quality of the manuscript.

Reviewer #1

I found the reviewer responses to be well done and all of my comments were addressed.

We appreciate the reviewer's kind comment.

Reviewer #2

Thank you so much for your thoughtful and thorough response to my comments. Generally, I am satisfied with your changes and appreciate the additional citations and supplemental materials. Your section around the GO annotations is much improved and supportive of your conclusions. My only concern that remains revolves around my first question: why was m/z exclusion used instead of mass exclusion? Regardless of Dupre 2021, in order to compare your mass exclusion approach to the current state of the art, I would expect one to try the built-in mass exclusion list on the instrument (made all the better if Advanced Peak Determination, APD, is enabled). It is very possible that the conclusion of Dupre would come to pass, but I think the extra experiment is worthwhile.

We thank the reviewer one more time for this comment. However, we observed that the mass exclusion scheme alone does not bring much performance boost in complex samples. The proteoform ID boost by FLASHIda is observed only when the accurate deconvolution by FLASHDeconv is coupled with quality-based exclusion as demonstrated in Supplementary Figure 1. In our preliminary results in which the mass exclusion (instead of quality-based) scheme was used in FLASHIda, the number of proteoform IDs was even slightly lower than that of standard acquisition with m/z exclusion, although the mass exclusion in FLASHIda yielded lots of distinct precursor masses as expected (30-40% more masses than the current FLASHIda). The main reason for this low ID count was the low quality of acquired precursor mass signals. Indeed, we developed the quality scheme (QScore and TQScore) to address this

phenomenon. When we applied the developed quality-based exclusion, the number of distinct precursor masses reduced by 30-40% but the overall proteoform level ID count increased by >100%. This shows that the simple mass exclusion may not be an optimal scheme for top down proteomics analysis of complex samples, as already presented by Dupre et al., 2021.

Moreover, in our standard acquisition used to generate ST datasets, the advanced peak determination (APD) was enabled during our LC-MS runs, thus the precursor charges are assigned by APD. But we found that only ~40% precursors of all MS2 spectra have assigned charges by APD; for instance, only 4,394 out of 10,105 precursors in ST90' dataset had the APD assigned precursor charges. Therefore, not to mention the accuracy of the assigned charges, if we apply the mass exclusion coupled with APD, it would be the exclusion scheme only with a small subset (~40%) of all precursors with charge assignment (a precursor mass cannot be determined without the charge assignment). Considering the low performance of the mass exclusion coupled with FLASHDeconv, which assigns masses to >80% of precursors real-time, APD based mass exclusion would not bring significant performance boost as compared with standard m/z exclusion. We thus believe that an additional experiment with an APD-based mass exclusion scheme for complex datasets would not strengthen the current manuscript.

Reviewer #3

The authors have carefully addressed all issues or suggestions and also provided a detailed file to explain in details how to deal with those issues.

I agree with the authors' technical improvement on their manuscript.

Only one point or advice on Page 22 of the rebuttal file which is how to improve the identification of the chimera spectra. The current version of FLASHIda uses a precursor (targeted) SNR to control the acquisition. If there are two proteoforms within one isolation window, then they will be co-fragmented. Although they are difficult to be identified when compared to species with a higher precursor SNR, they are still possible to be identified if we can identify those two precursors, such as identifying them by their corresponding isotopic envelopes. Furthermore, the proportion of chimera spectra is very high in real-life complex samples. So my concern or suggestion is we should consider these proteoforms when it is required to determine their acquisition possibility. I hope maybe in the next version, the authors

can further improve the performance of FLASHida by increasing the number of IDs from this type of proteoforms.

This is indeed a great idea for a future research direction. The identification of chimeric spectra is algorithmically challenging as the reviewer stated. But our main concern regarding chimeric spectra lies not on the algorithmic challenges but on the difficulty in FDR control. Given that non-chimeric spectrum identification results in high FDR values uncontrolled by conventional target decoy approach (as demonstrated in Methods, Supplementary Fig. 5-6 and Supplementary Table. 1), allowing the acquisition of chimeric spectra could even worsen the FDR estimation accuracy. Thus, with the absence of a proper FDR estimation or control method, the current purpose of FLASHida acquisition is to minimize the interference level (or maximize precursor SNR) given the selected mass subject to fragmentation. In the future, when a proper FDR control (that takes both chimeric and non-chimeric spectra into account) is established, FLASHida could be enhanced so it allows the acquisition of chimeric spectra. The identification of acquired chimeric spectra is not the focus of FLASHida (as this is an acquisition method) and should be dealt with by dedicated identification algorithms. We added the following paragraph in the Discussion section (page 18).

“Therefore, the current version of FLASHida focuses on obtaining precursors with low interference to minimize precursor errors as well as to avoid coeluted precursors (using precursor SNR). However, with respect to coeluted precursors, FLASHida would allow acquisition of chimeric spectra from them if an appropriate precursor mass error estimation method were developed. Dedicated algorithms for the identification of chimeric spectra could then increase the identification rate further.”